# Nongenetic surface engineering of mesenchymal stromal cells with polyvalent antibodies to enhance targeting efficiency

Tenghui Ye[1,5], Xi Liu[1,5], Xianghua Zhong[1], Ran Yan[1] & Peng Shi ®[1,2,3,4] ✉

Systemic infusion is a prevalent administration method for mesenchymal stromal cells (MSCs) in clinical trials. However, the inability to deliver a large number of therapeutic cells to diseased tissue is a substantial barrier. Here, we demonstrate that surface engineering of MSCs with polyvalent antibodies can effectively improve the targeting efficiency of MSCs to diseased tissue. The polyvalent antibody is directly synthesized on the cell surface via DNA template-directed biomolecule assembly. The data show that engineered MSCs exhibit superior adhesion to inflamed endothelium in vitro and in vivo. In female mouse models of acute inflammation and inflammatory bowel disease, engineered MSCs show enhanced targeting efficiency and therapeutic efficacy in damaged tissues. Notably, the entire procedure for polyvalent functionalization only requires the simple mixing of cells and solutions under physiological conditions within a few hours, which significantly reduces preparation processes and manufacturing costs and minimizes the impact on the cells. Thus, our study provides a strategy for improved MSC-based regenerative medicine.

Mesenchymal stromal cells (MSCs) have become a widely used type of therapeutic cell in the field of regenerative medicine due to their immunomodulatory, anti-inflammatory, and proangiogenic properties[1–3]. Transplanted MSCs exert their therapeutic effects by improving the microenvironment of damaged tissues through the paracrine pathway, showing extremely high application value in the treatment of various diseases, such as ischemic myocardial injury, acute liver injury, and inflammatory bowel disease (IBD)[4–7]. However, as is common in the field of stem cell therapy, the inability to deliver a large number of therapeutic cells to diseased tissue with significant efficiency is a substantial barrier to the effective application of MSC therapies[8,9]. Systemic infusion is a prevalent administration method for MSCs due to its repeatable nature and minimal invasiveness[10,11], and this method was used in 43% of the 914 MSC clinical trials from 2004–2018 included in Clinical Trials[12]. Unfortunately, typically about 1% of infused MSCs reach the target tissue[13,14]. This low targeting efficiency severely limits the clinical efficacy of MSCs[15,16].

Studies have shown that when MSCs reach blood vessels in damaged tissue, they adhere to the activated endothelium through the interactions between cell surface ligands and the overexpressed receptors on vascular endothelial cells and then migrate across the endothelial layer[17–19]. However, culture-expanded MSCs exhibit heterogeneous marker expression, resulting in low adhesion efficiency[20]. To overcome the problem of low adhesion efficiency and enhance the targeting capacity of MSCs, researchers have explored methods such as enzymatic glycosylation[21,22], covalent bonding[23,24], and hydrophobic insertion[25,26] to modify adhesion molecules on MSC surfaces and improve the adhesion efficiency of modified MSCs to damage sites.

[1]School of Biomedical Sciences and Engineering, South China University of Technology, Guangzhou International Campus, Guangzhou 511442, PR China. [2]National Engineering Research Center for Tissue Restoration and Reconstruction, South China University of Technology, Guangzhou 510006, PR China. [3]Guangdong Provincial Key Laboratory of Biomedical Engineering, South China University of Technology, Guangzhou 510006, PR China. [4]Key Laboratory of Biomedical Materials and Engineering of the Ministry of Education, South China University of Technology, Guangzhou 510006, PR China. [5]These authors contributed equally: Tenghui Ye, Xi Liu. ✉e-mail: pxs301@scut.edu.cn

These nongenetic engineering strategies can significantly reduce preparation processes and manufacturing costs and have received increasing attention in recent years[27–29]. However, these functionalization methods are primarily focused on monovalent functionalization, which is the display of single biomolecules across the cell surface. The limitation of these modifications is the low affinity delivered by the monovalent interaction, resulting in reduced adhesion of the modified MSCs under hemodynamic conditions, which has not yet been effectively improved.

Cell adhesion is achieved through polyvalent interactions[30]. Inspired by the fact that polyvalent interactions are superior to monovalent interactions[31,32], we envision that the construction of polyvalent molecules on the cell surface could be an effective way to improve the adhesion efficiency of MSCs to injured tissues. As vascular cell adhesion molecule 1 (VCAM1)-mediated MSC arrest on the endothelium is critical for effective MSC homing, in this study, we modify polyvalent anti-VCAM1 on the MSC surface to enhance the adhesion of intravenously infused MSCs to injured tissue vessels. We covalently conjugate DNA to anti-VCAM1 and construct polyvalent anti-VCAM1 using DNA self-assembly techniques. The polyvalent antibody is applied to the surface of MSCs through engineering. The resulting engineered MSCs exhibit increased adhesion to vascular endothelial cells in vitro and in vivo and show enhanced therapeutic efficacy in IBD mice. We demonstrate that this engineering approach enhances the targeting of MSCs to inflammatory tissues without affecting their proliferation, attachment, or paracrine functions and has excellent biosafety properties in mice. These results indicate that polyvalent-engineered MSCs are promising for disease treatment.

## Results

### Synthesis of polyvalent anti-VCAM1 (PAV) via DNA template-directed antibody assembly

In this study, we used DNA hybridization chain reaction (HCR) technology to prepare polyvalent antibodies[33]. The system includes a DNA initiator (DI) and two DNA monomers (DM); DM1 and DM2 (Supplementary Fig. 1). The DI sequentially opens the DNA monomers one by one and initiates polymerization to form a DNA polymer (Fig. 1a). The DNA monomer was designed with an amino attached to the end to connect to the antibody. Electrophoresis gel images demonstrated that the DNA polymers could be synthesized by DM1 and DM2 via the HCR in the presence of DI (Fig. 1a). Then, we covalently connected DM1 and DM2 to anti-VCAM1 (Fig. 1b). Electrophoretic gel images showed that the bands of synthesized DNA-anti-VCAM1 were higher than those of anti-VCAM1, indicating the increased molecular weight of anti-VCAM1. The UV–vis absorption spectra revealed that DNA-anti-VCAM1 had a significantly higher absorbance than anti-VCAM1 at 260 nm and 280 nm. These results demonstrated the ligation of anti-VCAM1 to DNA. We further examined whether binding to DNA affected the function of the anti-VCAM1 antibody. We used two cell lines, the endothelial cell line C166 that overexpresses VCAM1 and the control cell line K562 that does not express VCAM1. Flow cytometry showed that the anti-VCAM1 protein was able to bind to C166 but not K562 cells with or without DNA ligation, suggesting that DNA ligation did not alter the specificity of anti-VCAM1 (Fig. 1c). To polymerize DNA-anti-VCAM1 monomers into polyvalent structures, DNA-anti-VCAM1 monomers and DI were incubated for 3 h in a neutral buffer. Electrophoresis gel images revealed that DNA-anti-VCAM1 monomers could form protein multimers with larger molecular weights in the presence of DI (Fig. 1d). Overall, these data suggest that polyvalent biomolecules can be efficiently and systematically constructed by DNA self-assembly under physiological conditions.

### Construction of PAV on the cell surface

We used a lipid-DNA-directed bottom-up self-assembly strategy to construct PAV on the cell surface. As shown in Fig. 2a, the DI with cholesterol at the end was inserted into the cell membrane by hydrophobic intercalation, thus anchoring the oligonucleotide on the cell

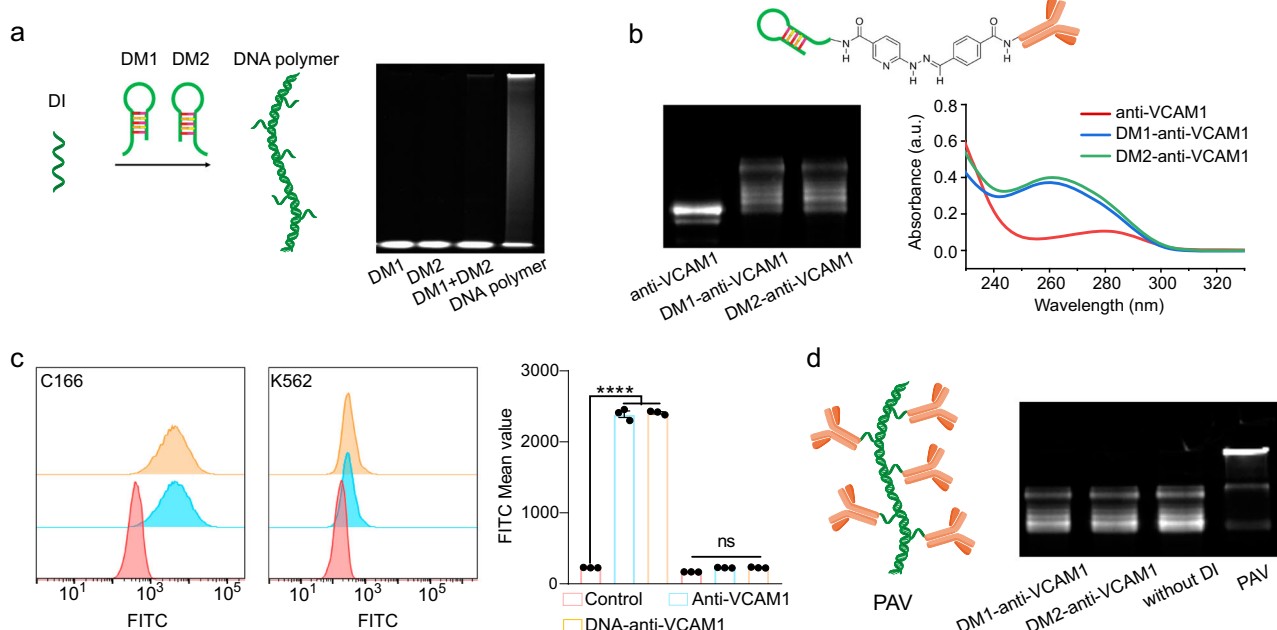

**Fig. 1 | Synthesis and characterization of PAV. a** Schematic illustration of DI-initiated HCR of DNA to form a DNA polymer (left) and electrophoresis gel image of the DNA polymer (right). **b** Schematic diagram of DNA-anti-VCAM1 and characterization of DNA-anti-VCAM1 with electrophoresis gel image (left) and UV–vis absorption spectra (right). **c** Flow cytometric analysis of the fluorescence intensity of VCAM1 and DNA-anti-VCAM1 binding on K562 and C166 cells. ****$P < 0.0001$, ns: $P = 0.3719$. Mean ± SEM, $n = 3$ independent replicates. **d** Schematic diagram of PAV (left) and characterization of PAV with electrophoresis gel imaging (right). For **a**, **b** and **d**, experiments were repeated three times independently with similar results obtained. Statistical analysis was performed by one-way ANOVA with Tukey's multiple comparisons tests (**$P < 0.01$; ***$P < 0.001$; ****$P < 0.0001$; NS, nonsignificant). Source data are provided as a Source Data file.

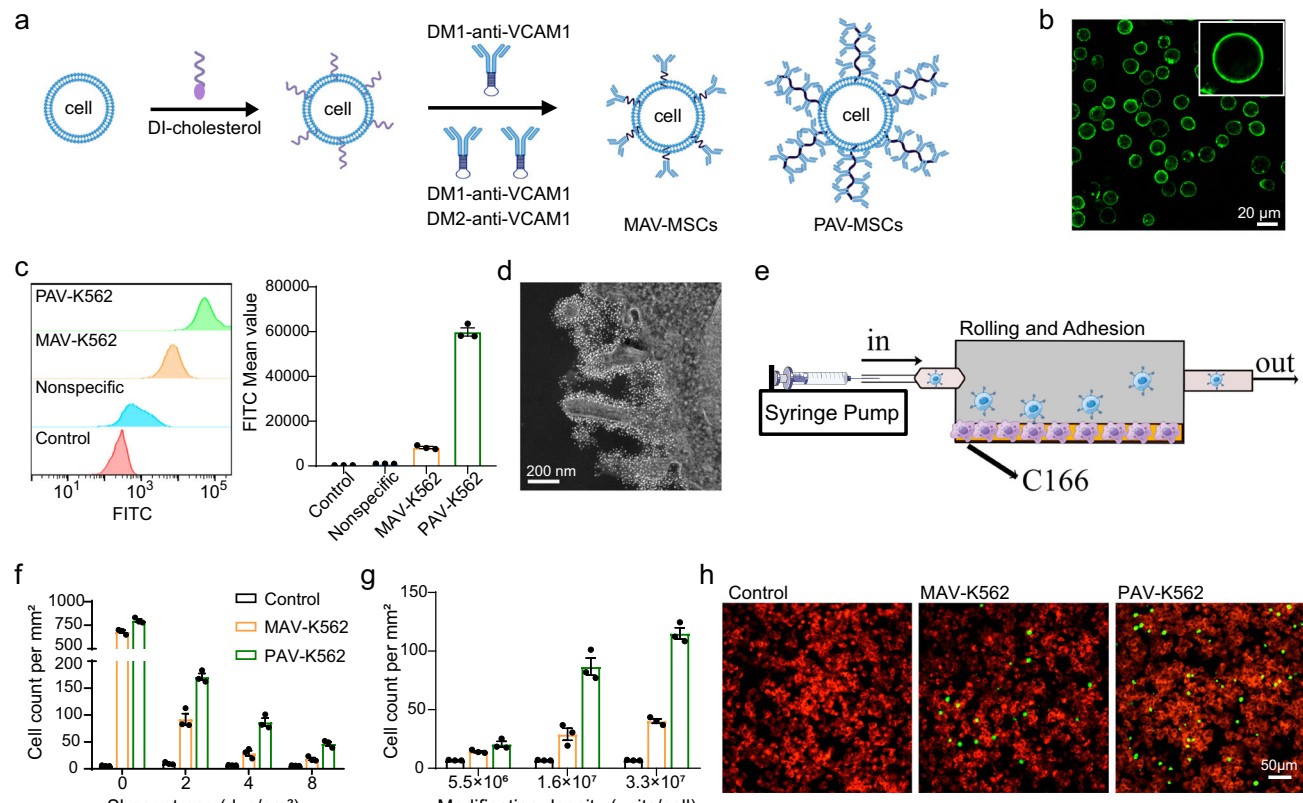

**Fig. 2 | Enhanced adhesion of PAV-engineered K562 cells to vascular endo-thelial cells. a** Schematic diagram of the bottom-up assembly of MAV or PAV on the cell membrane. **b** Confocal fluorescence image of PAV-engineered K562 cells. **c** Flow cytometric analysis of the fluorescence intensity of K562 cells modified with MAV or PAV. Mean ± SEM, $n = 3$ independent replicates. **d** Representative STEM image of polyvalent engineered K562 cells. Bright spots: approximately 10 nm quantum dots. **e** Schematic illustration showing the rolling and adhesion of engi-neered K562 cells on C166 cells under flow conditions. **f**, **g** Number of engineered K562 cells adhering to C166 cells. K562 cells were modified with MAV or PAV at a density of $1.6 \times 10^7$ units/cell and tested under different shear stress conditions (**f**). K562 cells were modified with different densities of MAV or PAV on the cell surface and tested at the same shear stress condition of 4 dyn/cm² (**g**). Mean ± SEM. $n = 3$ independent replicates. **h** Representative images showing engineered K562 cells (green) adhering to C166 cells (red). Shear stress: 4 dyn/cm². Representative images out of 7 images obtained are shown. Source data are provided as a Source Data file.

surface. PAV-engineered cells were then prepared by simultaneously incubating DI-modified cells with DM1-anti-VCAM1 and DM2-anti-VCAM1. As a control group, monovalent anti-VCAM1 (MAV)-engi-neered cells were also prepared by incubating DI-modified cells with DM1-anti-VCAM1 alone.

We first performed a series of experiments to screen for pairs of DNA sequences that, when formed into multimers on the cell surface, could load sufficient antibodies. In addition to DNA sequence 1 (Fig. 1), we synthesized four different sets of DNA sequences. All five sets of DNA monomers were able to self-assemble into DNA polymers after the addition of DI (Supple-mentary Fig. 2). The DNA monomers in each set were individually conjugated with FITC-labeled IgG to form DNA-IgG monomers. Then, monovalent or polyvalent IgG was formed on the surface of K562 cells by the bottom-up approach. The number of IgG molecules in the polyvalent structure formed by different sets of DNA sequences could be quantified by determining the fluores-cence intensity of FITC. After comparing the fluorescence inten-sities of the five sets of DNA sequences by flow analysis, it was found that DNA sequence 1 could load as many as 8 IgG proteins per DNA scaffold on the cell surface. Thus, we used the first set of DNA sequences to construct PAV on the cell surface for the fol-lowing experiments.

To better evaluate the adhesion of the engineered cells to vascular endothelial cells, we used the suspension cell line K562 as the model. Thus, the primary driving force for cell adhesion would be specific recognition but not the natural state of cell adhesion. After

engineering K562 cells with PAV using the method described above, the cells were first characterized using confocal fluorescence micro-scopy, and the fluorescence images showed that the surface of the K562 cells exhibited an intense green fluorescence signal (Fig. 2b). Further quantification of the fluorescence signal on the surface of K562 cells by flow cytometry showed an 8-fold increase in the fluorescence intensity in the PAV group compared to the MAV group (Fig. 2c and Supplementary Fig. 3), indicating an average of 8 anti-VCAM1 mole-cules assembled on each DNA scaffold. We also quantified the density of PAV on the cell surface by measuring the density of DI, since each DI generates one DNA scaffold. For example, at 0.5 μM DI, the density of PAV was -1.6 × 10⁷ units/cell (Supplementary Table 1). To better view the structure of the polyvalent antibody on the cell surface, we replaced the protein with similarly sized quantum dots (QDs) and used the same method to assemble DNA-QDs into multimers on the cell surface. Scanning transmission electron microscopy (STEM) images clearly revealed the extension of the multimeric structure of QDs from the cell surface, suggesting that the DNA assembled monomers into multimers through self-assembly (Fig. 2d). We then examined the stability of both monovalent and polyvalent surface modifications. Following a 24 h exposure of engineered cells to serum-containing culture medium, PAV-engineered cells maintained a surface antibody level of nearly 20%, corresponding to 2.4 × 10⁷ antibodies. While MAV-engineered cells exhibited a surface antibody level of less than 10%, corresponding to 0.15 × 10⁷ antibodies on the surface (Supplementary Fig. 4a, b). Notably, the data highlights a staggering 16-fold increase in the remaining surface antibodies on PAV-engineered cells relative to

MAV-engineered cells. We also conducted an experiment to investigate the potential for protein exchange between the engineered cells and neighboring native cells. The results demonstrated that no antibodies were detected on the surface of native cells following coculture with engineered cells, indicating that protein exchange is not occurring between the engineered cells and neighboring cells (Supplementary Fig. 4c). This finding suggests that the engineered cells do not affect other cells in their vicinity.

### Enhanced adhesion of PAV-engineered K562 cells to vascular endothelial cells

To determine whether surface engineering with PAV could mediate and enhance the adhesion of K562 cells to vascular endothelial cells, we incubated engineered K562 cells with C166 cells under static conditions. Fluorescence microscopy images showed that MAV and PAV could mediate the adhesion of K562 cells to C166 cells. As expected, the number of adherent K562 cells in the PAV group was much higher than that in the MAV group (Supplementary Fig. 5).

Considering the in vivo shear stress conditions generated by blood flow in the vasculature[34], a cell adhesion assay under static conditions does not accurately mimic the adhesion of engineered cells to vascular endothelial layers in vivo[35]. Therefore, we designed a flow adhesion experiment to study cell adhesion under well-defined shear conditions (Fig. 2e). In this experiment, a precision syringe pump combined with a well-defined flow chamber (µ-Slide I Luer) was used to provide stable shear stress conditions (0–8 dyn/cm$^2$). C166 vascular endothelial cells were seeded in the flow chamber, and native or engineered K562 cells rolled and adhered to C166 cells under defined shear stress. First, K562 cells were modified with MAV or PAV at a density of $1.6 \times 10^7$ units/cell, and different shear stress conditions were used in the cell adhesion assay. The results revealed that the number of adherent K562 cells in the PAV group was higher than that in the MAV group under all shear stress conditions (Fig. 2f). As the shear stress increased from 2 dyn/cm$^2$ to 4 dyn/cm$^2$, the ratio of the number of adherent cells in the PAV group to the MAV group increased from 2- to 3-fold, suggesting that polyvalent engineered K562 cells had better adhesion at higher shear stress. Under the highest shear stress condition of 8 dyn/cm$^2$, MAV mediated negligible cell adhesion. In comparison, PAV-engineered cells could adhere to C166 cells. We then fixed the shear stress at 4 dyn/cm$^2$ and gradually increased the density of MAV or PAV on the cell surface (Fig. 2g, h). Consistent with the previous results, the number of adherent K562 cells in the PAV group was higher than that in the MAV group for all density conditions. Collectively, we demonstrated the successful construction of PAV on the cell surface using a DNA-templated protein assembly strategy, and this polyvalent engineering technology significantly enhanced the adhesion of the modified cells to vascular endothelial cells under static and dynamic conditions.

### Adhesion and migration of engineered MSCs under shear stress conditions

Encouraged by the compelling results from experiments on K562 cells, we proceeded to produce engineered MSCs via the same strategy. Confocal fluorescence images showed strong FITC signals localized on the cell membrane (Fig. 3c). Flow cytometry indicated an 8-fold increase in the fluorescence intensity in the PAV group compared to the MAV group (Fig. 3b). These results demonstrated the successful preparation of PAV-engineered MSCs.

To study the adhesion and transendothelial migration of engineered MSCs under flow conditions, we used a 3D flow chamber to simulate an in vivo-like blood vessel. The bottom of the chamber was lined with collagen I containing SDF-1α inside, and C166 cells were cultured on the gel matrix to generate an activated endothelial monolayer (Fig. 3a). The microfluidic device mimicked a blood vessel with inflammation. Then, MSCs were modified with MAV or PAV at a density of $1.6 \times 10^7$ units/cell and perfused into the flow chamber under defined shear stress conditions. As shown in Fig. 3d, MAV and PAV engineering enhanced the adhesion of MSCs to target C166 cells. We specifically quantified the increase in the efficiency of cell adhesion mediated by engineered MSCs compared to native MSCs (Fig. 3e). Under a shear stress of 2 dyn/cm$^2$, cell engineering with MAV enhanced the adhesion efficiency by 230%, whereas the enhancement efficiency in the PAV group was 560%. When the shear stress was increased from 2 dyn/cm$^2$ to 8 dyn/cm$^2$, the ratio of enhancement efficiency in the PAV group to the MAV group further increased from 2.4- to 3.8-fold. The sharp increase in the enhancement efficiency suggests that polyvalence can lead to a higher degree of cell adhesion than monovalence, especially under higher shear stress conditions. We also varied the modification density of MAV or PAV on the cell surface and compared the difference between monovalent and polyvalent engineering on MSC adhesion. As expected, cell engineering with PAV resulted in a higher adhesion efficiency for all modification density conditions (Supplementary Fig. 6 and Fig. 3f).

After demonstrating the successful adhesion of MSCs to C166 cells under shear flow conditions, we investigated whether these MSCs migrated toward SDF-1α in the chamber. Confocal microscopy images were taken at 0 h and 24 h to track the migration of MSCs in the 3D flow chamber. It was clear from the 3D imaging results that native and engineered MSCs could migrate from the endothelial layer into the gel matrix at 24 h (Fig. 3g). Quantification analysis revealed that the number of migrating cells was higher in the PAV group than in the control and MAV groups (Fig. 3h, i). These results suggest that surface engineering does not affect MSC migration across the vascular endothelium and that polyvalent engineering increases the number of migrating cells by enhancing adhesion efficiency. We also found that most of the antibodies modified on the engineered cells were shed during migration (Supplementary Fig. 7). Taken together, above results demonstrate that surface engineering of MSCs with PAV holds great potential for enhancing cell adhesion to the endothelial layer and transendothelial migration under a physiological shear–stress range in vascular microcirculation.

### Effect of surface engineering on MSC functions and in vivo biosafety

Before proceeding to the in vivo study, we first investigated whether cell surface engineering affected MSC functions. MSCs were modified with MAV or PAV at densities of $1.6 \times 10^7$ and $2.4 \times 10^7$ units/cell, and we tested cell viability, proliferation, adhesion, and paracrine effects. The results of live/dead staining showed that native and engineered MSCs exhibited a steady growth state with almost no signs of cell death (Supplementary Fig. 8a). The CCK-8 assay indicated that engineered MSCs exhibited comparable proliferation potential relative to native MSCs (Supplementary Fig. 8b). In addition, engineered MSCs could effectively adhere to tissue culture plastic substrate with no abnormalities in morphology (Supplementary Fig. 8c). Increasing evidence indicates that MSCs exert their beneficial effects mainly through the secretion of factors. There are several paracrine cytokines released by MSCs that are involved in tissue repair, such as vascular endothelial growth factor (VEGF), fibroblast growth factor (FGF), interleukin 6 (IL-6), stromal cell-derived factor 1 (SDF-1), monocyte chemoattractant protein 1 (MCP-1), transforming growth factor-beta (TGF-β1), and hepatocyte growth factor (HGF)[36]. Therefore, we further tested the paracrine effect using ELISA. As shown in Supplementary Fig. 9a, in comparison to that of native MSCs, the expression level of several paracrine cytokines in engineered MSCs showed no discernable variation after 72 h of culture, which indicated that cell surface engineering did not affect the secretory behaviors of MSCs. Taken together, our results revealed that the essential function of MSCs was not considerably changed by cell surface engineering.

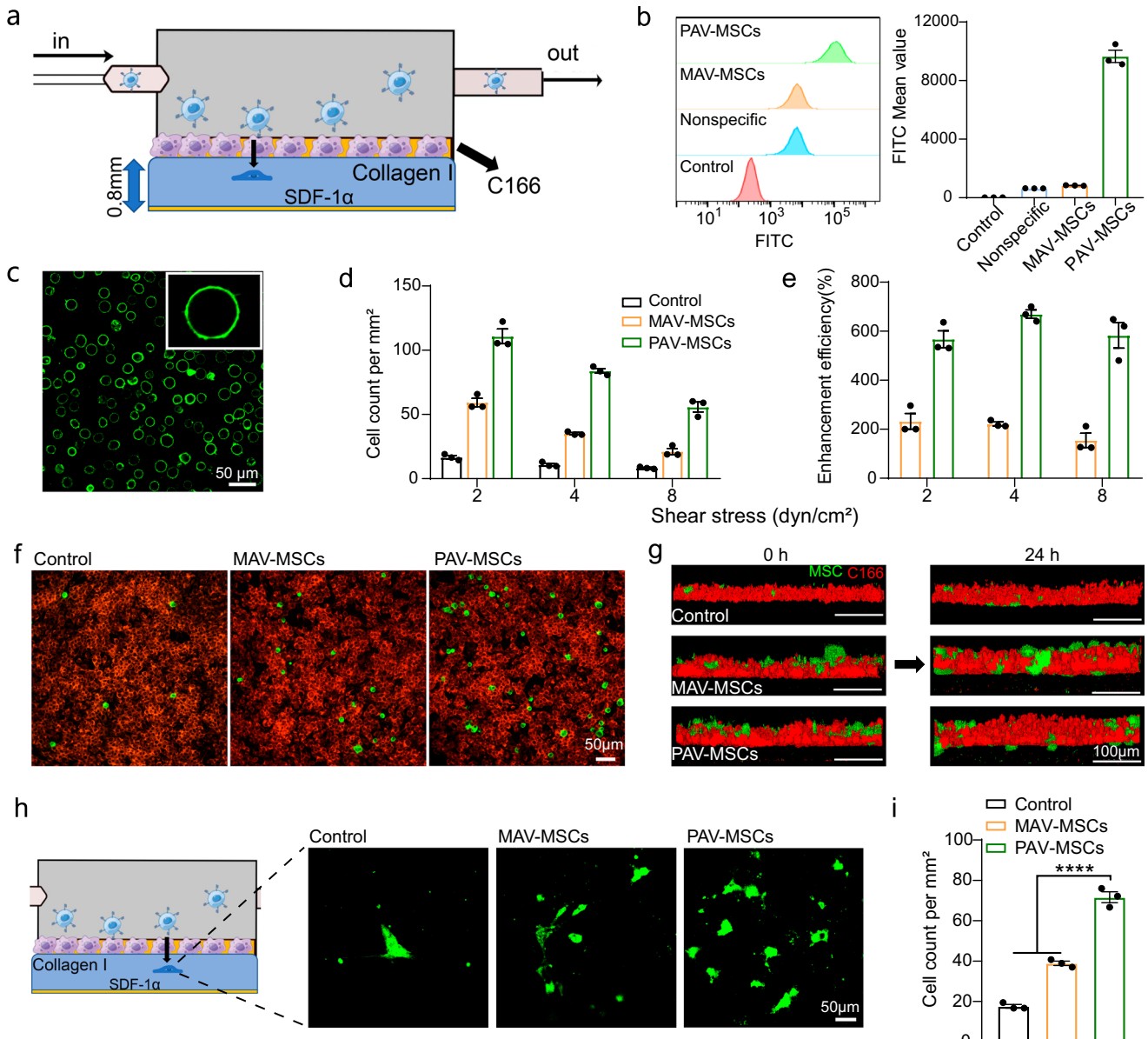

**Fig. 3 | Adhesion and migration of engineered MSCs under shear stress conditions. a** Schematic diagram of the experimental setup showing MSC adhesion and migration in a flow chamber. **b** Flow cytometric analysis of the fluorescence intensity of MSCs modified with MAV or PAV. Mean ± SEM, $n = 3$ independent replicates. **c** Confocal fluorescence image of PAV-engineered MSCs. Representative images out of 7 images obtained are shown. **d** Number of engineered MSCs adhering to C166 cells. MSCs were modified with MAV or PAV at a density of $1.6 \times 10^7$ units/cell, and tested under different shear stress conditions. **e** Quantitation of the enhanced efficiency of cell adhesion mediated by MAV and PAV. For **d**, **e**, Mean ± SEM, $n = 3$ independent replicates. **f** Representative images showing

engineered MSCs (green) adhering to C166 cells (red). Shear stress: 4 dyn/cm². Representative images out of 7 images obtained are shown. **g** Three-dimensional confocal fluorescence imaging of engineered MSC (green) migration across C166 cell layers (red) at 0 and 24 h. **h** Representative images showing the migration of MSCs into collagen I. Representative images out of 7 images obtained are shown. **i** Quantitation of the number of MSCs undergoing migration. ****$P < 0.0001$. Mean ± SEM, $n = 3$ independent replicates. Statistical analysis was performed by one-way ANOVA with Tukey's multiple comparisons tests (**$P < 0.01$; ***$P < 0.001$; ****$P < 0.0001$; NS, nonsignificant). Source data are provided as a Source Data file.

Considering that DNA may induce an adaptive immune response[37,38], we further tested the immunogenicity of the engineered MSCs in vivo. We collected whole blood and spleens from mice after intravenously administering engineered MSCs and analyzed T-cell activation and cytokine release. As shown in Supplementary Fig. 9b–f, neither T-cell activation nor inflammatory cytokine release was observed, suggesting that these engineered MSCs did not induce adaptive immune responses in vivo. Next, the biosafety of the engineered MSCs was further assessed after 14 days of treatment. The results showed that mouse body weight was not abnormal within 14 days. In addition, histological

examination showed that MSC treatment did not induce any toxic effects or tissue damage (Supplementary Fig. 10). The major organs, including the heart, liver, spleen, lungs, and kidneys, did not show visible damage, as indicated by H&E-stained sections. These results demonstrate that the engineered MSCs possess excellent biosafety in mice.

**In vivo targeting properties of engineered MSCs**
After demonstrating the in vivo biosafety of engineered MSCs, we evaluated whether PAV engineering could promote more efficient delivery of MSCs to sites of inflammation in vivo. We established a

mouse model of acute local ear inflammation by intradermally injecting bacterial-derived lipopolysaccharide into the pinna and used saline in the contralateral ear as a control[39]. This design enabled the quantitative comparison of MSC delivery efficiency between inflamed and noninflamed contralateral tissue within the same mouse (Fig. 4a)[40]. The successful establishment of the mouse model was verified by immunofluorescence staining, which showed that VCAM1 expression was significantly upregulated in the blood vessels of the inflamed ear compared to the control ear (Supplementary Fig. 11).

MSCs were labeled with Vybrant-DiD and administered intravenously to the mice 6 h after LPS injection, and IVIS imaging was subsequently performed at four time points to track MSC trafficking to inflamed tissue. Inflamed and control ears from the same animal were imaged in pairs to allow direct comparisons. From 12 h to 72 h after administration, PAV-engineered MSCs showed a dramatic increase in homing to the inflamed ear compared to native or MAV-engineered MSCs (Fig. 4b). The highest fluorescence intensity in the PAV group occurred at 48 h after administration, which was 3.5-fold higher than that in the MAV group (Supplementary Fig. 12). We also used intravital confocal microscopy to image the vasculature in the inflamed ear. PAV-engineered MSCs effectively adhered to inflamed blood vessels (Supplementary Fig. 13), suggesting that the improved homing efficiency was mediated by enhanced cell adhesion to the activated endothelium. To obtain more quantitative information, we performed flow cytometric analysis of cells collected from mouse ears. The results showed

that 5.4 % of PAV-engineered MSCs eventually reached the target tissue, which was 3.2 times higher than that in the MAV group and 6.6 times higher than that in the native group (Fig. 4c), which was consistent with the IVIS imaging results. In addition, there were no differences in the numbers of native and engineered MSCs localized within the noninflamed ear, suggesting that PAV-mediated MSC homing is specific. We compared the difference between MAV and PAV engineering on MSC homing efficiency. Engineering with MAV and PAV enhanced the homing efficiency of cells by 110% and 560%, respectively (Fig. 4d). These quantitative data demonstrated that cell surface engineering with multivalent anti-VCAM1 enabled MSCs to be specifically and efficiently delivered to inflamed tissue in vivo. Finally, we used immunofluorescence staining to examine whether the engineered MSCs could extravasate across the endothelium into surrounding tissues (Fig. 4e). Fluorescence images revealed that the majority of engineered MSCs were located outside the vessel lumen or colocalized with the endothelial layers, indicating that engineered MSCs retained the ability of transendothelial migration in vivo.

We subsequently evaluated the in vivo biodistribution of engineered MSCs, which demonstrated a predominant accumulation in the lung, liver and spleen, with minimal presence noted in the heart and kidney (Supplementary Fig. 14a, b). One obstacle for intravenous administration of therapeutic cells is that the cells can get trapped in the lung capillaries[41]. Interestingly, IVIS imaging and flow cytometry showed reduced entrapment of engineered MSCs in the lung compared to native MSCs (Supplementary Fig. 14c). Although the specific

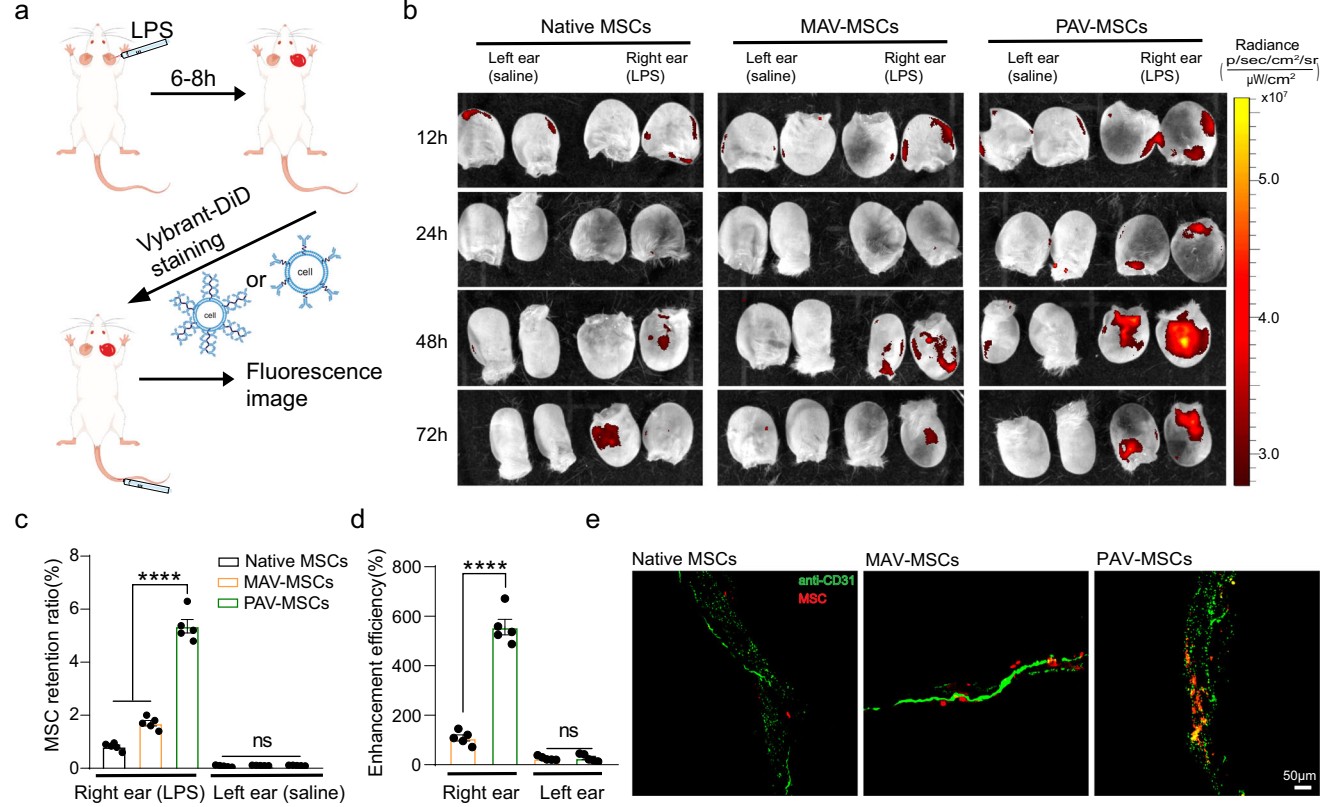

**Fig. 4 | The in vivo targeting properties of engineered MSCs. a** Schematic diagram of the in vivo experiments. Female BALB/c mice were injected with LPS subcutaneously to induce acute local ear inflammation. **b** IVIS imaging of control and inflamed ears from animals that received native or engineered MSCs. The mice were treated according to the method described in a and then administered native or engineered MSCs. The mice were euthanized at the indicated time points after administration, and the ears were immediately removed for imaging with an IVIS Lumina Series III. $n = 5$ mice. **c** Percentage of injected MSCs retained at the ears. MSCs were collected from mouse ears at 48 h after administration. ****$P < 0.0001$,

ns: $P > 0.9999$. **d** Quantitation of the enhanced homing efficiency in the MAV and PAV groups. ****$P < 0.0001$, ns: $P = 0.8282$. For **c**, **d** Mean ± SEM., $n = 5$ mice. **e** Immunofluorescence staining of the mouse ear to analyze MSC transendothelial migration. Mouse ears were collected, and whole mounts were stained with anti-mouse CD31 (green). Representative images out of 7 images obtained are shown. Statistical analysis was performed by one-way ANOVA with Tukey's multiple comparisons tests (**$P < 0.01$; ***$P < 0.001$; ****$P < 0.0001$; NS, nonsignificant). Source data are provided as a Source Data file.

mechanism is not clear, the reduction in lung entrapment contributed to the improvement in MSC homing efficiency. In addition, it should be emphasized that while MAV- and PAV-engineered MSCs had similarly low levels of lung entrapment, PAV-engineered MSCs showed 3.2 times more homing efficiency to the inflamed ear than MAV-engineered MSCs, suggesting that polyvalent engineering enhanced MSC homing to inflammation sites in vivo.

### Therapeutic efficacy of engineered MSCs against IBD

We further tested the therapeutic efficacy of engineered MSCs in a disease model of IBD. Various studies have shown that transplanted MSCs exert their therapeutic effects on IBD through immunomodulation and angiogenesis and represent a promising alternative treatment[42,43]. Here, we established a dextran sulfate sodium (DSS)-induced colitis model to test the ability of engineered MSCs to provide enhanced therapeutic benefits in mice with colitis[44]. As shown in Fig. 5a, normal C57 mice were given 3.5% DSS for 7 consecutive days after one week of acclimatization to create a colitis model. On Day 7, MSCs were intravenously administered. Additionally, we included control groups that received PBS, MAV, or PAV treatment[45]. We first examined the adhesion of engineered MSCs in the colons of mice at 48 h after intravenous administration. IVIS images and microscopic observations of tissue sections indicated that, compared with native and MAV-engineered MSCs, PAV-engineered MSCs showed the strongest targeting capacity to bowel tissue (Fig. 5b, Supplementary Fig. 15). Quantitative analysis showed that the homing efficiency for PAV-engineered MSCs was 4.7% (Fig. 5c), which was 3.5 times higher than that in the MAV-MSCs group and 6.0 times higher than that in the native group. Biodistribution assays in mice with colitis, consistent with findings in mouse model of acute ear inflammation, indicated that the engineered MSCs were less entrapped in the lung compared to native MSCs (Supplementary Fig. 16).

Then, we evaluated the therapeutic effects of engineered MSCs by monitoring the body weight and disease activity index (DAI, including the consistency index, weight loss index, and fecal bleeding index) of each group of mice. Compared to mice in the normal group, mice in the PBS group showed obvious weight loss and sustained increases in DAI scores, indicating the successful establishment of the colitis model. However, mice that received treatment showed varying degrees of improvements in the disease, indicating the gradual restoration of intestinal function after treatment. Mice in the PAV-MSCs group exhibited superior weight recovery and lower DAI scores compared to other groups, with statistically significant differences observed between the PAV-MSCs and MAV-MSCs groups (Fig. 5d, e). We also examined colon length, which is an essential index to evaluate the treatment efficacy of colitis in mice. As shown in Fig. 5f, g, mice treated with PAV-engineered MSCs had significantly longer colons than those in other treatment groups, suggesting that PAV-engineered MSCs effectively promoted the repair of damaged bowel tissue.

In addition, various inflammation-related mediators in bowel tissue have been measured to better understand the biological mechanism of engineered MSCs. Myeloperoxidase (MPO) activity, which correlates with the level of neutrophil infiltration in the colon[46], was significantly elevated in the PBS group and was suppressed to varying degrees after treatment. In particular, MPO activity in the PAV-MSCs group was decreased close to the level of the normal group (Fig. 5h). Moreover, PAV-MSCs group exhibited a statistically significant decrease in the levels of proinflammatory cytokines such as IL-6 and TNF-α, and a statistically significant increase in the level of the anti-inflammatory cytokine IL-10 compared to MAV-MSCs group (Fig. 5i–k). We then examined histological sections. H&E staining revealed inflammatory infiltration, mucosal destruction, and structural disruption of crypt foci in colitis mice in the PBS group. However, mice treated with PAV-engineered MSCs showed virtually normal pathological structures, suggesting that PAV-engineered MSCs ameliorated histological damage caused by DSS (Fig. 5l).

Notably, we found that treatment with antibodies alone led to a modest reduction in colitis severity, with no discernible variance in therapeutic outcomes between PAV and MAV (Fig. 5d–l). By contrast, PAV-MSCs could promote tissue repair more effectively. These results suggest that the primary therapeutic impact within the PAV-MSCs construct can be attributed to the MSC component, and the strategic incorporation of PAV modification on the MSC surface enhance the delivery of MSCs to damaged tissues, consequently yielding a more potent therapeutic effect. The above assessment further highlights the pivotal role of surface modification in bolstering the overall therapeutic efficacy of PAV-MSCs. Furthermore, the enhanced targeting efficiency and therapeutic efficacy of PAV-engineered MSCs were demonstrated through comparison with several engineered MSCs previously reported in the literature for intravenous injection into mice with colitis (Supplementary Table 3). Taken together, the in vivo results demonstrate that cell surface engineering with PAV can enhance the therapeutic effect of MSCs by enhancing MSC adhesion to damaged tissue and has the potential for significant applications in a wider range of diseases.

## Discussion

The loss of key ligands during MSC expansion in vitro leads to poor homing efficiency after systemic injection, which severely limits the therapeutic efficacy of MSCs[20,21]. Studies have shown that engineering MSCs with adhesion ligands could be an effective way to improve the targeting efficiency of MSCs following systemic infusion[27–29]. When performing cell surface engineering, it would be ideal to modify cells with the least amount of exogenous biomolecules on the surface while achieving the desired functions. Based on this principle, our polyvalent engineering strategy shows obvious advantages over monovalent engineering strategies. First, only a small amount of biomolecule is tethered on the cell surface via hydrophobic interactions, which ensures that the original composition of the cell membrane will not be significantly altered. Moreover, as each initiator grows into a supramolecule with a multivalent structure, sufficient functional groups will be displayed on the cell surface to enhance molecular recognition. In addition, PAV is approximately 50–100 nm in length and protrudes from the cell surface, as indicated in the STEM image. The extension of PAV will minimize the effect of nonspecific interactions and steric hindrance from the cell surface, ensuring molecular recognition. Our results demonstrated that PAV-engineered MSCs exhibited excellent targeting to damaged tissues, indicating great potential for clinical applications.

Compared with genetic engineering methods, the entire procedure for polyvalent functionalization only requires the simple mixing of cells and solutions under physiological conditions for a few hours (Supplementary Table 4), which significantly reduces preparation processes and manufacturing costs. We estimate that the cost of reagents and consumables for cell surface engineering is approximately $8 per million cells (Supplementary Table 5). The median dose administered intravenously is 100 million MSCs/patient/dose[12]. The average cost for preparing 100 million PAV-engineered MSCs is $800, demonstrating the advantages of our engineered approach to large-scale cell processing. Using the polyvalent engineering approach holds great promise for clinical applications, especially considering the present high costs associated with stem cell therapy[47]. In addition, nongenetic methods do not permanently alter cells at the genetic level, offering the possibility of temporarily manipulating cellular functions while avoiding potential safety risks[27]. We have shown that the polyvalent engineering approach preserves MSC functions, including proliferation and their ability to secrete paracrine factors. PAV-engineered MSCs showed great biosafety in vivo and significant therapeutic efficacy against damaged tissues. It should be noted that

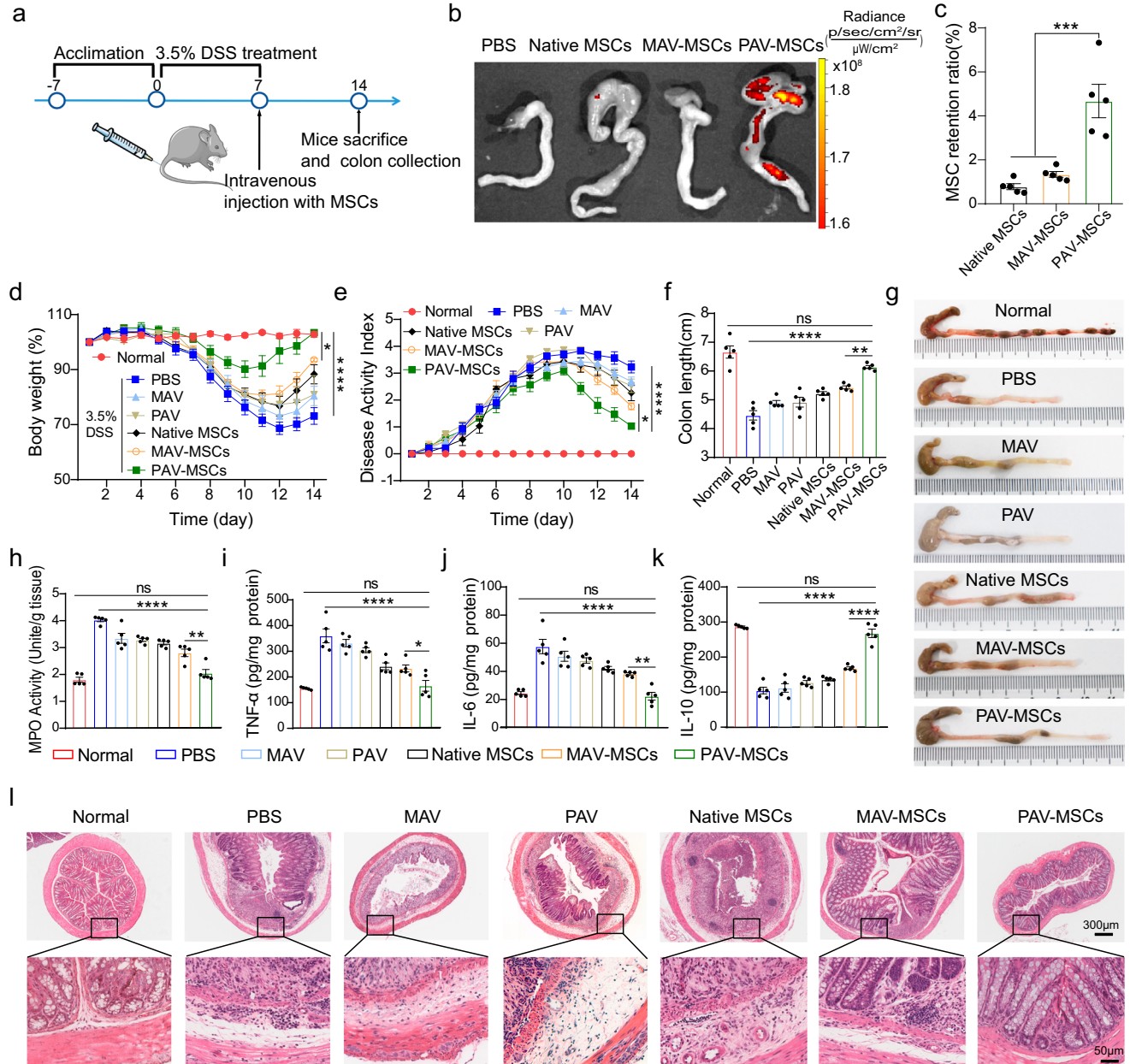

**Fig. 5 | The therapeutic efficacy of engineered MSCs in DSS-induced colitis.**
**a** Schematic diagram showing the treatment of mice with DSS-induced colitis with engineered MSCs. Female C57BL/6 mice were given drinking water containing 3.5% DSS from Day 0 to Day 7. The mice were intravenously injected with PBS or native or engineered MSCs on Day 7. After 7 days of treatment, the mice were sacrificed, and the colon was collected for further analysis. **b** IVIS imaging of colon tissues. The mice were euthanized at 48 h after administration, and the colon was immediately removed for imaging with an IVIS Lumina Series III. $n = 5$ mice. **c** Percentage of injected MSCs retained at the colon. MSCs were collected from colon at 48 h after administration. ***$P = 0.0001$. Mean ± SEM., $n = 5$ mice. **d** Changes in the body weight of mice receiving different the treatments within 14 days. *$P = 0.0108$, ****$P < 0.0001$. **e** DAI scores of mice in each group over 14 days. *$P = 0.0483$,

****$P < 0.0001$. Quantitative analysis of colon length (**f**) and the appearance of colons harvested from mice (**g**) after the different treatments. **$P = 0.0041$, ****$P < 0.0001$, ns: $P = 0.0545$. For **d**, **e** and **f**, Mean ± SEM, $n = 5$ mice. **h** The MPO activity of colons after the different treatments. **$P = 0.0024$, ****$P < 0.0001$, ns: $P = 0.7207$. **i**–**k** The levels of TNF-α, IL-6, and IL-10 in colon tissues after the different treatments. **i** *$P = 0.0459$, ****$P < 0.0001$, ns: $P = 0.9996$. **j** *$P = 0.0079$, ****$P < 0.0001$, ns: $P = 0.9933$. **k** ****$P < 0.0001$, ns: $P = 0.6006$. For **h**–**k**, Mean ± SEM, $n = 5$ mice. **l** Representative H&E staining images of colon tissue harvested on Day 14 after the different treatments. Representative images out of 7 images obtained are shown. Statistical analysis was performed by one-way ANOVA with Tukey's multiple comparisons tests (*$P < 0.5$; **$P < 0.01$; ***$P < 0.001$; ****$P < 0.0001$; NS, nonsignificant). Source data are provided as a Source Data file.

although we have emphasized the differences between genetic and nongenetic engineering methods, these two technologies can be integrated to achieve better therapeutic efficacy of MSCs. Because PAV engineering is independent of cell type, we provide a convenient tool that can be used for functionalizing native or genetically engineered cells.

It is also important to note that we propose a general approach for MSC engineering rather than simply preparing PAV-engineered

MSCs. Although we constructed linear DNA structures to direct antibody assembly, DNA can be designed to form different two- or three-dimensional structures[48,49]. Based on Watson-Crick base-pairing rules, the advanced structures formed by DNA self-assembly are predictable and programmable[50,51]. Thus, when conjugated with diverse biomolecules, these rationally designed DNA structures can be used as scaffolds to precisely control biomolecule assembly[52,53]. For example, we could design DNA structures modified with

adhesion ligands at well-defined positions and investigate the effect of ligand spatial distribution on molecular recognition in future studies. For another example, a bifunctional structure including sialyl Lewisx and anti-VCAM1 could be prepared. Engineering MSCs with such multimers allows for the exploration of the role of synergistic mechanisms in enhancing the adhesion capacity of MSCs. We propose that DNA assembly is a powerful tool for cell surface engineering[54].

In conclusion, we developed a simple and easy-to-perform method for surface engineering MSCs with polyvalent antibodies. The engineered MSCs exhibit enhanced targeting efficiency and therapeutic efficacy on damaged tissues, indicating promising potential for clinical applications. We hope our study will provide a new perspective for MSC-based regenerative medicine.

## Methods

### Ethical statement

All mice were treated according to the standards in the Guide for the Care and Use of Laboratory Animals. The South China University of Technology Animal Care and Use Committee authorized all animal operations used in our research (2022020).

### Animals

Female BALB/c mice and female C57BL/6 mice were bought from Hunan SJA Laboratory Animal Co., Ltd and housed in specified pathogen-free (SPF) animal facilities. All mice were maintained in pathogen-free settings in a room with a 12 h light/dark cycle and a regulated temperature (22 °C) and humidity (45–60%).

### Materials

Oligonucleotides (Supplementary Table 6) were synthesized by Sangon Biotech Co., Ltd (Shanghai, China). The oligonucleotide powder was fully dissolved in PBS solution to a concentration of 100 μM, annealed, and stored at −20 °C. Flow antibodies including FITC anti-mouse VCAM1 (catalog: 105706), Ultra- Ultra-LEAF™ anti-mouse VCAM1 (catalog: 105728), FITC anti-mouse CD31 (catalog: 102506), FITC anti-mouse CD3 (catalog: 100204), Percp anti-mouse CD4 (catalog: 100434), Alexa Fluor® 700 anti-mouse CD8 (catalog: 100730), APC anti-mouse CD69 (catalog: 104514), PE anti-mouse CD25 (catalog: 101903) were purchased from Biolegend. Conjugation of DNA-antibody was performed using SoluLink® bioconjugation technology. S-4FB Crosslinker was purchased from Solulink (catalog: S-1004-010), S-HyNic Crosslinker was purchased from Solulink (catalog: S-1002-105), Turbolink Catalyst Buffer was purchased from Solulink (catalog: S-2006-105). Murine SDF-1alpha (catalog: 250-20 A) was purchased from PeproTech. Cell Counting Kit-8, DiO/DiD cell-labeling solution, and Live/Dead viability/cytotoxicity kit were purchased from Beyotime Biotechnology Co., Ltd. (Shanghai, China). DMEM (high glucose and low glucose), RPMI 1640, trypsin-EDTA, and penicillin−streptomycin were purchased from Gibco (Life Technologies, Carlsbad, CA, US). Fetal bovine serum (FBS) was purchased from Biological Industries (catalog: 04-001ACS).

### Cell culture

Vascular endothelial cells C166 (ATCC, CRL-2581) were cultured in DMEM high glucose medium containing 10% fetal bovine serum, and chronic myelogenous leukemia K-562 cells (ATCC, CCL-243) were cultured in 1640 medium containing 10% fetal bovine serum supplemented with 1% penicillin and 1% streptomycin. Mouse primary MSCs were purchased from iCell Bioscience Inc (MIC-iCell-s018), cultured in DMEM low glucose medium containing 10% fetal bovine serum supplemented with 1% penicillin and 1% streptomycin. Cells were incubated at 37 °C in a 5% $CO_2$ and a 95% RH atmosphere. All experiments were performed using MSCs in passages number 3–6.

### Surface engineering of K562 cells/MSCs

$1 \times 10^6$ K562 cells or MSCs were resuspended with 1 mL of PBS. The cells were centrifuged at $300 \times g$ for 5 min and the supernatant was removed. The cells were washed again and resuspended in 200 μL of PBS solution. The DI-cholesterol solution was added to a final concentration of 0.5 μM and incubated with cells in a 600 rpm oscillating metal bath for 20 min to immobilize DNA initiators on the cell surface. At the end of the reaction, the cells were washed twice with 1 mL of PBS to remove excess DI-cholesterol and then resuspended in 200 μL of PBS. For monovalent engineering cells, 0.25 μM of DM1-protein was added to the cell suspension; for polyvalent engineering cells, both 0.25 μM of DM1-protein and DM2-protein were added to the cell suspension. The cell suspensions were incubated in a 400 rpm oscillating metal bath at room temperature for 3 h for HCR. Gently blow the cells with a pipette every hour to prevent them from sinking. The engineered cells were centrifuged and washed three times with PBS before being used for fluorescence imaging and flow cytometry analysis. MSCs were modified with MAV or PAV at densities of $1.6 \times 10^7$ (the concentration of DI-cholesterol for co-incubation is 0.6 μM) units/cell in vivo experiment.

### Quantitation of DI on the cell surface

$2 \times 10^6$ K562 cells or MSCs were suspended in PBS with different concentrations (0.05–2 μM) of DI-cholesterol and incubated at room temperature for 20 min. After washing three times with PBS, the cells were treated with DI-CS-FAM (0.05–2 μM), which is complementary to DI, and incubated at room temperature for 1 h. After washing and counting, the cells were lysed by RIPA lysis buffer (Sangon, C500005-0100), and fluorescence was quantified by fluorescence spectrophotometer. With eliminating cellular autofluorescence, the readouts were fitted to a standard curve to determine DI numbers on the cell surface.

### The proliferation and paracrine of engineered MSCs

MSCs were modified with MAV or PAV at densities of $1.6 \times 10^7$ and $2.4 \times 10^7$ units/cell. For proliferation analysis, engineered MSCs were seeded in a 96-well plate at a density of 3,000 cells per well and cultured in 200 μL of complete medium. After 24 h or 48 h culture, 10 μL of CCK-8 reagent was added to each well and incubated for 1 h at 37 °C. Absorbance was measured at 450 nm using a Microplate Reader. Cells seeded in a 24-well plate were stained with a Live/Dead Double Staining kit and fluorescence images were captured using a Fluorescence Microscope at the appropriate time point. For paracrine analysis, MSCs were seeded in a 12-well plate at a density of $5 \times 10^4$ cells per well with complete medium for 1 day to allow the cells to fully adhere to the wall, and then incubated with serum-free medium for 3 days. The supernatant was collected by centrifugation (3000 rpm, 20 min) and assayed using ELISA kits according to the manufacturer's instructions. ELASA kits were purchased from Shanghai Enzyme-linked Biotechnology Co., Ltd.

### Cell rolling and adhesion under shear stress conditions

Rolling and adhesion assays were performed using a microfluidic device. We used a microfluidic chamber obtained from ibidi (Germany μ-Slide I 0.4 Luer, 80176). According to the manufacturer's instructions, C166 cells were seeded on the chamber to achieve a final density of $1 \times 10^5$ cells/cm² and grown in the chamber to 100% confluence. The channel length, width and height of the chamber are 50 mm, 5.0 mm and 400 μm respectively. The growth area is 2.5 cm² per channel and channel volume is 100 μL. After labeling K562 cells or MSCs with the fluorescent probe DiO according to the manufacturer's instructions, the cells were processed to surface engineering with MAV or PAV. One million cells were resuspended in 10 mL of PBS for each group. Precision syringe pumps are used to provide stable and controlled shear

stress. After preparing the equipment, the flow rates were adjusted according to the desired shear stress: 1 mL/min (2 dyn/cm²), 2 mL/min (4 dyn/cm²), 4 mL/min (8 dyn/cm²):

$$\tau = \eta \cdot 131.6 \cdot \Phi \qquad (1)$$

$\tau$ = shear stress (dyn/cm²), $\eta$ = dynamical viscosity (dyn s/cm²), $\Phi$ = flow rate (mL/min).

Rolling and adhesion activities of native or engineered cells were observed and recorded for further quantitative analysis. After the flow, the remaining cell suspension in the chamber was removed and fluorescence imaging was immediately performed. The cells on the μ-Slide were imaged using a Nikon fluorescence microscope, and the number of engineered cells adhering per field was quantified by counting green fluorescent cells with ImageJ.

### MSC adhesion and migration under shear stress conditions
We used 3D microfluidic chamber obtained from ibidi (Germany, μ-Slide I Luer 3D, 87176) to perform adhesion and migration assays. This chamber allows for the culture of adherent cells on a 3D gel matrix under flow conditions, facilitating transendothelial migration studies. The chamber well can be filled with gel and utilized for cell culture and microscopic analysis. The well dimensions are 5.4 mm × 4.0 mm with a height of 0.8 mm, resulting in a growth area per well of 0.21 cm². The bottom of the 3D flow chamber was lined with 2 μL of SDF-1α solution (500 ng/mL) and 15 μL of collagen I on top and incubated at 37 °C for 1 h to allow the collagen I to solidify. C166 cells were seeded on collagen I to achieve a final density of $1 \times 10^5$ cells/cm² and grown in the chamber to 100% confluence. One million native or engineered MSCs were suspended in 10 mL of PBS and perfused into the flow chamber under defined shear stress conditions. After cell adhesion assays under shear stress conditions, the PBS was changed to complete medium. Three-dimensional confocal fluorescence imaging was performed at the time point of 0 h and 24 h to image and quantify MSCs migrating into collagen I.

### Quantification of the enhanced efficiency of cell adhesion
For in vitro studies, the number of MSCs adhering to C166 cells were counted using ImageJ software. For in vivo studies, FlowJo was used to calculate the number of cells adhering to damaged tissue per group. The increase in the efficiency of cell adhesion mediated by engineered MSCs compared to native MSCs was calculated using the following equations:

$$\text{Enhancement efficiency}(\%) = (Nm(p) - Nc)/Nc \times 100 \qquad (2)$$

where Nm, Np, and Nc denote the number of cells adhering in the MAV-MSCs Group, PAV-MSCs Group and native MSCs Group, respectively.

### Biosafety analysis in vivo
6−8 weeks female BAL/B /c mice were randomized into four groups and received intravenous injection of $1 \times 10^6$ native or engineered MSCs. After 48 h, the mice were euthanized and whole blood and spleen were collected. For blood collection, whole blood was collected through orbital blood collection into a centrifuge tube filled with 20 μL of 0.5 M EDTA solution and centrifuged at $2,000 \times g$ for 15 min at 4 °C. The supernatant plasma was used to test the cytokine content by ELISA. The remaining cells were resuspended with 1 mL of PBS and treated with 10 mL of 1× RBC Lysis Buffer (Beyotime, C3702) at 4 °C for 10 min. After adding 10 mL of PBS to stop the reaction, remaining cells were collected by centrifugation ($450 \times g$, 5 min) for further flow analysis. For splenocyte collection, the spleen was excised and placed in a cell culture dish. The spleen was crushed with the syringe handle after the addition of 4 mL of pre-chilled PBS. The triturated solution was filtered through a 200-mesh nylon mesh into a 15 mL centrifuge tube and centrifuged at $450 \times g$ for 5 min at 4 °C. After treating with 2 mL of 1× RBC lysis buffer at 4 °C for 10 min, splenocytes were collected by centrifugation ($450 \times g$, 5 min) and resuspended in 2 mL PBS. The collected whole blood cells and splenocytes were stained with antibodies for flow cytometric detection.

### Evaluation of in vivo targeting of engineered MSCs
The LPS-induced dermal inflammation in the mouse pinna was established. Briefly, female BALB/c mice (6-8 weeks, average body weight: 19−20 g) were injected with 30 μg LPS (Sigma Aldrich, L6529, in 30 μL saline) into the posterior/dorsal dermis of the right ear, while the contralateral ear was injected with 0.9% saline as a control. After 6 h, $1 \times 10^6$ MSCs (native, MAV or PAV) suspended in 150 μL PBS were administered intravenously to the mice. MSCs were pretreated with Vybrant-DiD staining buffer according to the manufacturer's instructions.

**IVIS imaging of the ears**. Mice were euthanized at the specified time after intravenous injection of MSCs to obtain mouse ears. Fluorescence imaging was performed using IVIS Lumina Series III (PerkinElmer). Image analysis and quantification of fluorescence intensity were performed using Living Image 4.4.

**Flow cytometry analysis**. Animals were euthanized 48 h after intravenous injection of MSCs. The ear was removed at the level of the base, and the dorsal and ventral skin was peeled off and placed in 2 mL digestion buffer. After incubation for 1 h at 37 °C, the ear skin was pestled into a 70 μm cell strainer (Biosharp, BS-70-XBS). Strainers were washed with 2 mL 1% FBS and 2 mM EDTA in PBS. The cells were then treated with 1×RBC lysis buffer (Beyotime, C3702) for 2 min at 4 °C, washed with PBS and analyzed using a BD/FACS Celesta flow cytometer.

**Confocal imaging**. Intravital imaging of the vasculature in the inflamed ear was performed on laser confocal microscope (LSM 880 WITH AIRYSCAN) at 48 h after intravenous MSCs injection. For delineation of the vasculature during imaging, 100 μL of 10 mg/mL FITC-dextran (2000 kDa) was injected intravenously 2 h before imaging.

**Frozen section and immunofluorescence staining**. Mice were euthanized 48 h after receiving native or engineered MSCs intravenously. The ear was immediately removed and the hair was shaved from the ears. The ears were treated with 4% paraformaldehyde fixative for 6 h and washed with PBS to remove paraformaldehyde. The ears were then placed in a 30% sucrose solution and dehydrated overnight. They were then embedded at low temperature with optimal cutting temperature compound (OCT, SAKURA) and slicing with a freezing microtome (Leica, CM1950). The tissues were then incubated with dye-coupled primary antibodies diluted in primary antibody diluent overnight at 4 °C according to the manufacturer's instructions. Finally, the sections were stained with DAPI and observed under a fluorescence microscope.

**Biodistribution assay**. To induce acute ear inflammation, the mice were first administered subcutaneous injections of LPS as described above. $1 \times 10^6$ MSCs were stained with Vybrant-DiD and then injected intravenously into the mice. After 48 h, the mice were sacrificed and their organs were removed for fluorescence imaging using the IVIS Lumina Series III. The obtained images were analyzed and fluorescence intensity were quantified using Living Image 4.4

### Evaluation of therapeutic efficacy of engineered MSCs against IBD
Female C57BL/6 mice (8–12 weeks old, average body weight 19–22 g) were given 3.5% DSS for 7 consecutive days after one week of

acclimatization to establish the DSS-induced colitis model. DSS was purchased from Yeasen Biotechnology Co., Ltd (Shanghai, 60316ES25). Then, mice were randomly divided into four groups and treated with PBS (150 μL) or native/engineered MSCs (1 × 10⁶ cells in 150 μL PBS) intravenously. Throughout the treatment, the weight of the mice was recorded every day to monitor the treatment's effects. Based on the pretreatment weight, the percentage of weight loss is calculated. Mice were fasted for 6 h at the end of the experiment and then euthanized. The colon was photographed and its length was measured. The distal colon tissues were collected for various analysis, including H&E staining. ELISA was used to measure MPO activity and detect inflammatory cytokines.

**Disease activity assay.** Disease activity assay. For IBD mice, stool viscosity and occult blood levels were tested daily. Parameters representing the DAI were recorded. The DAI is calculated by combining the scores from the percentage of weight loss, fecal viscosity, and fecal occult blood. Histology assay. The distal colons of the mice were collected and fixed in 4% paraformaldehyde for 24 h. Colon tissues were then sectioned, embedded in paraffin, and examined under a light microscope after staining with H&E.

**Histology assay.** The distal colons of the mice were collected and fixed in 4% paraformaldehyde for 24 h. Colon tissues were then sectioned, embedded in paraffin, and examined under a light microscope after staining with H&E.

**Flow cytometry analysis.** The entire colonic tissue was excised with removing any attached connective tissue. The tissue was cut into small segments (0.5-1 cm long) and cleaned with washing solution (RPMI-1640 supplemented with 1% FBS). The tissue fragments were then transferred to a centrifuge tube and added with 10 ml of preheated HBSS buffer containing 5 mM EDTA at 37 °C. The tissue fragments were incubated at 37 °C in a shaking incubator for 40 min at 220 rpm, followed by discarding the supernatant and transferring the remaining tissue to RPMI-1640 containing type IV collagenase (1 mg/ml) and DNase I (0.1 mg/ml). After incubating at 120 rpm in a shaking incubator for 1 h, the suspension was passed through a 100-mesh strainer to remove any undigested tissue and then centrifuged at 300 × g for 5 min at 4 °C. Isolated cells were used for flow cytometry analysis

**Statistical analysis**
Data analyses were performed using GraphPad Prism8. Data were analyzed using the unpaired Student's t-test, one-way or two-way analysis of variance (ANOVA) to evaluate the statistical significance of group differences. $P < 0.05$ was considered to be statistically significant.

**Statistics & reproducibility**
No statistical methods were used to determine sample size. Data were presented as the means ± SEM. No data were excluded from the analyses. Randomization was used for animal studies. The Investigators were not blinded to allocation during experiments and outcome assessment.

**Reporting summary**
Further information on research design is available in the Nature Portfolio Reporting Summary linked to this article.

## Data availability
The data that support this study are available within the article, Supplementary Information or Source data files. Source data are provided with this paper.

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

## Acknowledgements

This work was supported by National Key R&D Program of China (2022YFB3808300 to P.S.), the National Natural Science Foundation of China (22277030 to P.S.), and the Natural Science Foundation of Guangdong Province (2021A1515012353 to P.S.).

## Author contributions

T.Y. performed the experiments, interpreted the data, and wrote the manuscript. X.L. did animal studies. X.Z. and R.Y. did experiments. P.S. conceived the concept, designed the study, interpreted the data, and wrote the manuscript.

## Competing interests

The authors declare no competing interests.

## Additional information

**Peer review information** *Nature Communications* thanks Oren Levy and Carston Wagner for their contribution to the peer review this work. A peer review file is available.

