## [Peer Review File · Nature Communications]

Reviewers' comments:

Reviewer #1 (Remarks to the Author):

In their manuscript titled: “Nongenetic surface engineering of mesenchymal stromal cells with polyvalent antibodies to enhance targeting efficiency” authors use DNA template-directed assembly of polyvalent anti-VCAM1 antibody to engineer the surface of MSCs as a strategy to improve their targeting to disease site. The cell engineering approach seems elegant, and the manuscript is well structured.

Nevertheless, MSC surface engineering via different approaches (genetic and non-genetic) as an efficient disease-targeting strategy was reported in the past decade by multiple groups, also resulting in improved clinical outcomes in pre-clinical disease models (some referenced in this manuscript). Moreover, anti-VCAM1 antibody was specifically used by multiple groups to coat MSCs for colon targeting in a colitis model (Ref 16 in this manuscript (Ko et al.), also see: *Med Sci Monit.* 2019; 25: 4457–4468. Chen et al. “VCAM1 Antibody-Coated Mesenchymal Stromal Cells Attenuate Experimental Colitis via Immunomodulation”). Accordingly, the novelty of this study is rather limited - while the antibody coating approach taken in this study uses a different coating protocol, it is unclear whether it holds a meaningful advantage over previously reported approaches (or results in significantly different targeting efficiency or clinical impact). Unfortunately, it is unlikely this yet additional cell surface engineering approach has a true potential to shift the thus far unsuccessful clinical translation fate of systemically infused MSCs.

Additional comments:

- An important control group is missing from the in-vivo studies (Fig. 6) – i.e., authors should have included treatment of mice with Anti-VCAM1 antibody alone (or liposomes incorporating anti-VCAM1 on their surface using the same DNA-guided approach). This is especially important since anti-VCAM1 antibody itself was reported to improve clinical outcomes in the same colitis model (for example, see: *Lab Invest.* 2000 Oct;80(10):1541-51. doi: 10.1038/labinvest.3780164. Soriano et al. “VCAM-1, but not ICAM-1 or MAdCAM-1, immunoblockade ameliorates DSS-induced colitis in mice”)
- Authors are encouraged to add statistics to Figure 3I.
- The statistical significance noted in Fig.6 C (body weight) and D (disease activity) is unclear – is PAV-MSCs statistically significant vs. MAV-MSCs and potentially indicate that polyvalency provide a meaningful clinical advantage in this model? Authors should clarify this in the text and in the figure as well (PAV-MSCs is significant vs. which groups?).
- Clarifying statistical analysis is also needed for Fig.6 E, 6 G & 6I – unclear from the figure or text which groups show significance (especially PAV-MSC vs MAV-MSC).

- Authors state “We estimate that the cost of reagents and consumables for cell surface engineering is approximately \$8 per million cells, which is affordable for therapeutic purposes”. However, no further details are provided. Since this is viewed by the authors as a strength for this approach, it would be helpful if they can provide a detailed breakdown of this estimate (and maybe even a comparison to costs of other cell engineering approaches referenced) – especially in the context of large-scale cell processing which is needed for clinical translation.

Reviewer #2 (Remarks to the Author):

In this manuscript the authors have designed a clever general approach for the non-genetic modification of cell surfaces with antibodies. The approach is inspired by previous work by others using DNA conjugated to a lipid, to functionalize cell membranes with DNA tags. In this report the authors use a DNA initiator conjugated to cholesterol, that after washing can be hybridized with an antibody that has been conjugated to a complementary DNA strand or multiple antibodies using a self-assembling DNA polymer approach. The polymer approach allows the assembly of multiple monospecific antibodies on a single strand or multiple antibodies with different specificities. The authors have chosen in this study to apply their approach to the targeting of MSCs to endothelial cells with anti-VCAM1 antibodies. The authors systematically demonstrate that they can modify the surface of MSCs. Through a series of well-conceived shear flow studies they show that polyvalent modified MSCs are able to bind tighter and accumulate to a significantly greater extent on vascular endothelium cells than either non-modified or monovalent modified MSCs. The authors go on to demonstrate that the modified cells can extravasate across a monolayer of endothelium cells. Most impressively, they carry out *in vivo* studies demonstrating that the polyvalent modified MSCs are able to accumulate to a much greater extent in regions of inflammation, which MSCs are attracted to, and with a colitis model show a significant therapeutic effect. No toxicity was observed from this approach. Taken together, this is rather inspiring report that not only lays out a unique cell surface modification approach, but more importantly, lays out a systematic *in vitro* and *in vivo* way of evaluating these kinds of approaches. While others may not choose to use the same cells or indications, this paper does an excellent job of laying out the core concepts that need to be considered when evaluating such an approach. I recommend acceptance, when the following criticisms have been addressed.

- 1) The stability of the surface modifications, both monovalent and polyvalent, is not addressed. What is the life-time? The modification only uses one cholesterol per modification potentially leading to exchange with other cells. It would be good to determine the extent of potential exchange with other close by cells and albumin.
- 2) Shedding of the modification as the cells move along the endothelium or through cells is not addressed. One would expect loss of the modification as the cells move through VCAM1 positive cells.

3) Only the distribution of the cells at the target site is quantitated. What about the biodistribution throughout the animal? What percentage of the injected cells accumulate at the sites of inflammation?

4) Fig. 4, since there is no change in the amounts of cells or cytokines, this figure could be moved into the SI.

5) Materials and Methods: More detail need so be added for reproducibility. Please go through each section. For example, what were the actual concentrations of the DI-cholesterol used should be added. When you say wash, what does that mean, specifically? Volume? How long? When you say fluorescence microscopy and quantitation, what fluorophore and what method of quantitation?

Reviewer #1:

In their manuscript titled: “Nongenetic surface engineering of mesenchymal stromal cells with polyvalent antibodies to enhance targeting efficiency” authors use DNA template-directed assembly of polyvalent anti-VCAM1 antibody to engineer the surface of MSCs as a strategy to improve their targeting to disease site. The cell engineering approach seems elegant, and the manuscript is well structured. Nevertheless, MSC surface engineering via different approaches (genetic and non-genetic) as an efficient disease-targeting strategy was reported in the past decade by multiple groups, also resulting in improved clinical outcomes in pre-clinical disease models (some referenced in this manuscript). Moreover, anti-VCAM1 antibody was specifically used by multiple groups to coat MSCs for colon targeting in a colitis model (Ref 16 in this manuscript (Ko et al.), also see: *Med Sci Monit.* 2019; 25: 4457–4468. Chen et al. “VCAM1 Antibody-Coated Mesenchymal Stromal Cells Attenuate Experimental Colitis via Immunomodulation”). Accordingly, the novelty of this study is rather limited – while the antibody coating approach taken in this study uses a different coating protocol, it is unclear whether it holds a meaningful advantage over previously reported approaches (or results in significantly different targeting efficiency or clinical impact). Unfortunately, it is unlikely this yet additional cell surface engineering approach has a true potential to shift the thus far unsuccessful clinical translation fate of systemically infused MSCs.

• Response: We appreciate the valuable feedback from the reviewer. These comments and suggestions have inspired us to conduct more in-depth studies to improve the manuscript. In this manuscript, we recognize the superiority of polyvalent interactions in cell adhesion and have developed an approach to modify polyvalent antibodies on the cell surface through DNA self-assembly. This polyvalent approach allows the assembly of multiple monospecific antibodies on a single strand or multiple antibodies with different specificities (as also noted by Reviewer #2). Thus, our approach of MSCs engineering involves more than just coating them with monovalent antibodies. In response to the concerns whether this approach improves the homing efficiency and therapeutic efficacy of MSCs, we have conducted the subsequent animal experiments and made point-by-point comparisons between previous approaches and our approach:

1. Previous studies have shown that the homing efficiency of native MSCs administered intravenously is generally less than 1% *in vivo*. (Zhang, Ming, et al. *FASEB* 21. 3197-3207 (2007); Sarkar, D. et al. *Blood* 118, e184-191 (2011).). To demonstrate the superior targeting efficacy of PAV-engineered MSCs, we quantified the percentage of MSCs successfully targeted and enriched in damaged tissues. Our results showed that PAV-engineered MSCs had a homing efficiency of 5.4% in the mouse ear inflammation model and 4.7% in the colitis model, as shown in Fig. 4c and 5c. Remarkably, this efficiency outperformed native MSCs by 6.6 times in the ear inflammation model and 6.0 times in the colitis model.

2. We compared the targeting efficiency of PAV-engineered MSCs with anti-VCAM1-coated MSCs which was previously reported for colitis treatment in mice. While the exact percentage of engineered MSCs targeting the colon was not directly quantified in those reports, a practical comparison can be inferred through the imaging and quantitative data presented in the article. The data included Fig. 2 from Ko IK, et al. *Mol Ther* 18, 1365-1372 (2010) and Fig. 6 from Chen, Qianqian, et al. *Med Sci Monit* 25, 4457-4468 (2019), with comparisons made to Fig. 5c and Supplementary Fig. 15 from our work. We conducted a comparison based on quantifying fluorescent cells in tissue sections and determining the fold change of engineered MSCs to native MSCs in the target organ. Our analysis yielded an inference that the targeting efficiency of previously reported engineered MSCs is approximately two-fold higher than native MSCs. This finding aligns with the results of our study showing the ratio of MAV-engineered MSCs to native MSCs, but falls short compared to PAV-engineered MSCs (6.0 times), emphasizing the considerable advantage of employing polyvalent engineering approach.

Fig. 4c. Percentage of injected MSCs retained at the ears. MSCs were collected from mouse ears at 48 h after administration. Mean \pm SEM (n=5).

Fig. 5c. Percentage of injected MSCs retained at the colon. MSCs were collected from mouse colon at 48 h after administration. Mean \pm SEM (n=5).

Supplementary Figure 15. Immunofluorescence staining of colon tissue. Immunofluorescence staining of the colon at 48 h post administration for analyzing the MSCs homing. Red: MSCs, Blue: DAPI.

3. We further conducted a comparison of the therapeutic effects of several engineered MSCs reported in the literature which were administered via intravenous injection to mice with colitis. Our analysis found that the utilization of PAV-engineered MSCs provided distinct advantages for the treatment of colitis in mice. Improvements were observed across all disease indicators, notably in body weight, colon length, and amelioration of histological damage (Supplementary Table 3).

References	Targeting efficiency	Therapeutic effects					
		Body weight	DAI	Colon length in colitis mice vs. normal mice	MPO activity in colitis mice vs. normal mice	Levels of TNF- α , IL-6, IL-10 in colitis mice vs. normal mice	Histological sections or pathological scores
PAV engineered-MSCs for colitis	4.67%	103%	1	0.92	1.1	1;0.91;0.93	virtually normal pathological structures
Anti-VCAM1-coated MSCs for colitis (Mol Ther 18, 1365-1372 (2010).)	NA	~95%	NA	~0.8	NA	NA	pathological scores 2.5
Anti-VCAM1-coated MSCs for colitis (Med Sci Monit 25, 4457-4468 (2019).)	NA	<95%	~1	NA	NA	NA	few inflammatory response
IFN- γ expressing MSCs by genetic engineering for colitis (Inflamm Res 64, 671-681 (2015).)	NA	~80%	~1.5	NA	NA	NA	pathological scores >1.5

IL1 β -pretreated MSCs for colitis (Cell Mol Immunol 9, 473-481 (2012).)	NA	~95%	~2.5	~0.8	NA	NA	pathological scores 2
ICAM-1 overexpressing MSCs by genetic engineering for colitis (Stem Cell Res Ther 10, 267 (2019).)	NA	~90%	NA	~0.85	NA	NA	pathological scores >2
SDF-1-pretreated MSCs for colitis (Stem Cell Res Ther 10, 204 (2019).)	NA	~95%	~1.5	~0.85	NA	~1; NA; ~1.3	hyperemia and edema were remitted; inflammatory cell infiltration was decreased
IFN- γ +IL1 β -pretreated MSCs for colitis (J Tissue Eng Regen Med 13, 1792-1804 (2019).)	NA	~90%	NA	~0.85	NA	NA	structural disruption of crypt foci
TLR-3-pretreated MSCs for colitis (Cytotherapy 18, 630-641 (2016).)	NA	~85%	~5	~0.86	NA	NA	pathological scores >2
Roe-inspired MSC microcapsules for colitis (Proc Natl Acad Sci U S A 118, e2112704118 (2021).)	NA	~95%	~2	~0.93	~1.5	~4; ~1.5; ~0.75	pathological scores >3.5
ASA-pretreated MSCs for colitis (Stem Cells Dev 23, 2093-2103 (2014).)	NA	NA	~5	NA	NA	NA; NA; ~2.5	infiltration of inflammatory cells; pathological scores >4

Supplementary Table 3. The comparison of PAV-engineered MSCs with previously reported engineered MSCs for the treatment of mice with colitis.

In summary, our findings suggest that PAV-engineered MSCs exhibit superior targeting efficiency and therapeutic efficacy. It is worth emphasizing that, as discussed in the manuscript, we propose a general approach to MSC engineering rather than simply preparing PAV-engineered MSCs. Considering the significant therapeutic potential of MSCs in treating various diseases, we anticipate that further advancements in DNA structures modified with adhesion ligands will boost the performance of engineered MSCs for various disease treatments. We have integrated above results and description into the revised manuscript.

1. An important control group is missing from the in-vivo studies (Fig. 6) – i.e., authors should have included treatment of mice with Anti-VCAM1 antibody alone (or liposomes incorporating anti-VCAM1 on their surface using the same DNA-guided approach). This is especially important since anti-VCAM1 antibody itself was reported to improve clinical outcomes in the same colitis model (for example, see: Lab Invest. 2000 Oct;80(10):1541-51. doi: 10.1038/labinvest.3780164. Soriano et al. "VCAM-1, but not ICAM-1 or MAdCAM-1, immunoblockade ameliorates DSS-induced colitis in mice".

• Response: We thank the reviewer for this great suggestion. As suggested, we conducted an experiment that included a control group receiving separate anti-VCAM1 treatment in mice with colitis. The quantitative data were added to Fig.5 (revised manuscript). Consistent with previous reports, we found that treatment with anti-VCAM1 antibodies alone attenuated colitis. Furthermore, the administration of engineered MSCs in mice resulted in a significantly enhanced therapeutic effect (Fig. 5d-5l). These results demonstrate the

superiority of anti-VCAM1 engineered MSCs therapy over traditional antibody treatment alone.

2. Authors are encouraged to add statistics to Figure 3I.

• Response: We thank the reviewer for this suggestion. The statistical analyses were added to Fig.3 in revised manuscript.

3. The statistical significance noted in Fig.6 C (body weight) and D (disease activity) is unclear – is PAV-MSCs statistically significant vs. MAV-MSCs and potentially indicate that polyvalency provide a meaningful clinical advantage in this model? Authors should clarify this in the text and in the figure as well (PAV-MSCs is significant vs. which groups?).

• Response: We appreciate the great suggestion from the reviewer. In the revised manuscript, we have included statistical analyses to demonstrate that the results presented in Fig. 5d and 5e (revised manuscript) for the PAV-MSC group compared to the MAV-MSC group (the main control group) are statistically significant. We have also described this result in the manuscript text.

.....Mice in the PAV group exhibited superior weight recovery and lower DAI scores compared to other groups, with statistically significant differences observed between the PAV and MAV groups.....

4. Clarifying statistical analysis is also needed for Fig.6 E, 6 G & 6I – unclear from the figure or text which groups show significance (especially PAV-MSC vs MAV-MSC).

• Response: We appreciate the great suggestion from the reviewer. In the revised manuscript, we have included a more specific statistical analysis in Fig. 5f and Fig. 5h-k (revised manuscript), which provides evidence of a significant treatment effect for the PAV-MSC group compared to the MAV-MSC group. We have also described this result in the manuscript text.

Fig. 5. d) Changes in the body weight of mice receiving different the treatments within 14 days. e) DAI scores of mice in each group over 14 days. f and g) Quantitative analysis of colon length (f) and the appearance of colons harvested from mice (g) after the different treatments. For d, e and f, Mean \pm SEM, n=5 mice. h) The MPO activity of colons after the different treatments. i-k) The levels of TNF- α , IL-6, and IL-10 in colon tissues after the different treatments. For h-k, Mean \pm SEM, n=5 mice. l) Representative H&E staining images of colon tissue harvested on Day 14 after the different treatments. Representative images out of 7 images

obtained are shown.

5. Authors state “We estimate that the cost of reagents and consumables for cell surface engineering is approximately \$8 per million cells, which is affordable for therapeutic purposes”. However, no further details are provided. Since this is viewed by the authors as a strength for this approach, it would be helpful if they can provide a detailed breakdown of this estimate (and maybe even a comparison to costs of other cell engineering approaches referenced) – especially in the context of large-scale cell processing which is needed for clinical translation.

• Response: We appreciate the great suggestion from the reviewer. As suggested by the reviewer, we have provided further details, including the procedures and timeline for the preparation of PAV-engineering cells (Supplementary Table 4) and the cost of preparing one million PAV-engineered cells (Supplementary Table 5).

Stem cell therapy costs can range anywhere from \$2500 - \$50,000 per treatment (Consumer Reports, January 11 (2018); BioInformant, April 1 (2023)). The median dose administered intravenously is 100 million MSCs/patient/dose. (Kabat, M., et al. Stem Cells Transl. Med.9, 17–27 (2020); Golpanian, S., et al. Stem Cells Transl Med 5,186-191(2015)). The average cost for preparing 100 million PAV-engineered MSCs is \$800, demonstrating the advantages of our engineered approach to large-scale cell processing. Combined with the superior targeting efficiency and low cost of polyvalent engineered MSCs, we propose that the approach of preparing polyvalent engineered cells holds great promise for clinical applications. We have integrated above results and description into the revised manuscript.

Cell engineering procedure	Time
DI co-incubation with cells	20 min
HCR	3 h
Centrifuging, washing	<30 min
All	<4 h

Supplementary Table 4. The procedures and timeline for the preparation of PAV-engineered cells.

Major reagents	Costs (\$)
DI-cholesterol (0.4 nmol)	0.6
DM1-antiVCAM1 (1.8 µg)	3
DM2-antiVCAM1 (1.8 µg)	3
Consumables	<0.5
All	8

Supplementary Table 5. The cost of preparing one million PAV-engineered cells.

Reviewer #2:

In this manuscript the authors have designed a clever general approach for the non-genetic modification of cell surfaces with antibodies. The approach is inspired by previous work by others using DNA conjugated to a lipid, to functionalize cell membranes with DNA tags. In this report the authors use a DNA initiator conjugated to cholesterol, that after washing can be hybridized with an antibody that has been conjugated to a complementary DNA strand or multiple antibodies using a self-assembling DNA polymer approach. The polymer approach allows the assembly of multiple monospecific antibodies on a single strand or multiple antibodies with different specificities. The authors have chosen in this study to apply their approach to the targeting of MSCs to endothelial cells with anti-VCAM1 antibodies. The authors systematically demonstrate that they can modify the surface of MSCs. Through a series of well-conceived shear flow studies they show that polyvalent modified MSCs are able to bind tighter and accumulate to a significantly greater extent on vascular endothelium cells than either non-modified or monovalent modified MSCs. The authors go on to demonstrate that the modified cells can extravasate across a monolayer of endothelium cells. Most impressively, they carry out in vivo studies demonstrating that the polyvalent modified MSCs are able to accumulate to a much greater extent in regions of inflammation, which MSCs are attracted to, and with a colitis model show a significant therapeutic effect. No toxicity was observed from this approach. Taken together, this is rather inspiring report that not only lays out a unique cell surface modification approach, but more importantly, lays out a systematic in vitro and in vivo way of evaluating these kinds of approaches. While others may not choose to use the same cells or indications, this paper does an excellent job of laying out the core concepts that need to be considered when evaluating such an approach. I recommend acceptance, when the following criticisms have been addressed.

• The reviewer provided a thorough and accurate description of our design and experiments. We greatly appreciate the positive comments as well as the constructive suggestions to help improve our manuscript.

1. The stability of the surface modifications, both monovalent and polyvalent, is not addressed. What is the life-time? The modification only uses one cholesterol per modification potentially leading to exchange with other cells. It would be good to determine the extent of potential exchange with other close by cells and albumin.

• Response: We thank the reviewer for this great suggestion. As suggested by the reviewer, we have examined the stability of both monovalent and polyvalent surface modifications (Supplementary Figure 4a, b). After exposing PAV-engineered cells to a culture with serum for 24 hours, the cells retained a surface antibody level of nearly 20%, corresponding to 2.4×10^7 antibodies on the surface. In comparison, MAV-engineered cells exhibited a surface antibody level of less than 10%, corresponding to 0.15×10^7 antibodies on the surface. Notably, the data highlights a staggering 16-fold increase in the remaining surface antibodies on PAV-engineered cells relative to MAV-engineered cells. Interestingly, we also observed that the presence of serum enhanced the stability of the surface modification.

Additionally, we conducted an experiment to investigate the potential for protein exchange between the engineered cells and neighboring native cells. The results demonstrated that no antibodies were detected on the surface of native cells following 3 h of co-culture with MAV- or PAV-engineered cells, indicating that protein exchange is not occurring between the engineered cells and neighboring cells (Supplementary Figure 4c). This is an encouraging result, as it indicates that the engineered cells do not affect other cells in their vicinity. It also suggests that there is a low risk of unintended immune reactions or other negative effects caused by the engineered cells. We have integrated above results and description into the revised manuscript.

Supplementary Figure 4. The stability analysis of the surface modifications. a and b) Residual levels of antibodies in MAV- or PAV-cells cultured in serum or serum-free medium. The cells were modified at a density of 1.6×10^7 DI units/cell. Following monovalent or polyvalent engineering, the cells were cultured in medium supplemented with either 10% serum or no serum. Residual fluorescence intensity on the cell surface was detected by flow cytometry at corresponding time points. Mean \pm SEM, $n=3$ independent experiments. c) Flow cytometric analysis of the fluorescence intensity of native cells following 3 h of co-culture with MAV- or PAV-cells.

2. Shedding of the modification as the cells move along the endothelium or through cells is not addressed. One would expect loss of the modification as the cells move through VCAM1 positive cells.

• Response: We thank the reviewer for pointing this out. As suggested by the reviewer, we quantified the fluorescence intensity of the engineered MSCs pre- and post-migration across the endothelium. As

anticipated, most of the antibodies modified on the engineered cells were shed during migration (Supplementary Figure 7).

Supplementary Figure 7. Shedding of antibodies after MSC migration. a) Schematic illustration of migration assay in transwell. b) Percentage of residual fluorescence intensity on engineered MSCs pre- (0 h) and post-migration (24 h). Mean \pm SEM, n=3 independent replicates.

3. Only the distribution of the cells at the target site is quantitated. What about the biodistribution throughout the animal? What percentage of the injected cells accumulate at the sites of inflammation?

• Response: We thank the reviewer for this great suggestion. We evaluated the biodistribution of engineered MSCs in mouse models of acute ear inflammation (Supplementary Figure 14) and colitis (Supplementary Figure 16), respectively. In mouse model of acute ear inflammation, the biodistribution assay of engineered MSCs displayed a predominant accumulation in the lung, liver, and spleen, with minimal presence noted in the heart and kidney. The results also displayed a reduced entrapment of engineered MSCs in the lung compared to native MSCs. Likewise, similar results emerged from the biodistribution assay of engineered MSC in mice with colitis.

Additionally, we quantified the percentage of engineered MSCs accumulated in the target tissues to assess their homing efficiency. In mouse model of acute ear inflammation, the results showed that 5.4% of PAV-engineered MSCs eventually reached the target tissue (Fig. 4c), which was 3.2 times higher than that in the MAV group and 6.6 times higher than that in the native group. In mouse model of colitis, the homing efficiency for PAV-engineered MSCs was 4.7% (Fig. 5c), which was 3.5 times higher than that in the MAV group and 6.0 times higher than that in the native group.

Supplementary Figure 14. Biodistribution of engineered MSCs in mouse model of acute ear inflammation. a) IVIS imaging of the mouse organs. b) Quantitation of fluorescence intensity of organs in different groups. Mice were euthanized at 48 h after intravenous injection of 1×10^6 DiD-labelled native or engineered MSC and the organs were removed for fluorescence imaging with IVIS Lumina Series III. Mean \pm SEM, n=5 mice.

Supplementary Figure 16. Biodistribution of engineered MSCs in mice with colitis. a) IVIS imaging of the mouse organs. b) Quantitation of fluorescence intensity of organs in different groups. Mice were euthanized at 48 h after intravenous injection of 1×10^6 DiD-labelled native or engineered MSCs and the organs were removed for fluorescence imaging with IVIS Lumina Series III. Mean \pm SEM, n=5 mice.

Fig. 4c. Percentage of injected MSCs retained at the ears. MSCs were collected from mouse ears at 48 h after administration. Mean \pm SEM (n=5).

Fig.5c. Percentage of injected MSCs retained at the colon. MSCs were collected from mouse colon at 48 h after administration. Mean \pm SEM (n=5).

4. Fig. 4, since there is no change in the amounts of cells or cytokines, this figure could be moved into the SI.

- Response: We thank the reviewer for the suggestion. As suggested, we have moved Fig. 4 to the supplementary file (Supplementary Figure 9).

5. Materials and Methods: More detail need so be added for reproducibility. Please go through each section. For example, what were the actual concentrations of the DI-cholesterol used should be added. When you say wash, what does that mean, specifically? Volume? How long? When you say fluorescence microscopy and quantitation, what fluorophore and what method of quantitation?

- Response: We thank the reviewer for pointing this out. As suggested by the reviewer, we have carefully reviewed the manuscript and added more details to the Materials and Methods section in order to improve reproducibility. The revised parts are highlighted in red in the updated manuscript.

REVIEWER COMMENTS

Reviewer #1 (Remarks to the Author):

The authors have done a nice work addressing some of the reviewers' comments and improving the level of the manuscript.

Especially interesting was Supplemental Table 3 comparing different engineered MSCs in a colitis model. While one should likely refrain from drawing conclusions from such comparisons of different experimental systems and conditions, it provides valuable insights. It mostly shows that the modest increase of number of MSCs reaching the disease site (authors report targeting efficiency of ~4.5%) is insufficient to create a real clinical advantage over other approaches - as evidenced by DAI, which was similar in other approaches, including a "simple" anti-VCAM coating (for instance the Med Sci Monit 2019 paper). Moreover, other engineering approaches, both genetic and enzymatic, have shown a significantly higher targeting efficiency in other models (the approach by Sackstein et al. Nat Met 2008 is only one example).

Addition of the Anti-VCAM1 antibody as a control group was important, strengthening previous reports that VCAM-1 antibody itself attenuates colitis (Revised Figure 5). Given the reported importance of PAV vs MAV reported by authors, it would have been appropriate to use such PAV antibodies for this control group, instead of a single monovalent antibody...

Moreover, the biodistribution experiments (important addition!) does not show a significant shift in biodistribution vs other engineering approaches, further highlighting a modest increase in targeting efficiency. The relative instability of this cell modification reported in Sup. Fig.4 and the antibody shedding reported in Sup. Fig.7 also raises concerns regarding this modification approach and its long-term clinical efficacy vs other approaches, such as stable/transient genetic modifications.

Overall, while the manuscript is now technically improved, one should wonder whether this report of yet another antibody coating approach provides meaningful clinical advantage or makes a substantial contribution to the fields of cell engineering or MSC therapies.

Reviewer #2 (Remarks to the Author):

The authors have addressed my criticisms. The manuscript is ready for publication.

Reviewer #1 (Remarks to the Author):

The authors have done a nice work addressing some of the reviewers' comments and improving the level of the manuscript.

Response: Thank the reviewer for recognizing our work. We are grateful for the opportunity to improve our manuscript based on the thoughtful feedback provided. And we hope that the following animal experiments and clarification address the reviewer's concerns satisfactorily.

Especially interesting was Supplemental Table 3 comparing different engineered MSCs in a colitis model. While one should likely refrain from drawing conclusions from such comparisons of different experimental systems and conditions, it provides valuable insights. It mostly shows that the modest increase of number of MSCs reaching the disease site (authors report targeting efficiency of ~4.5%) is insufficient to create a real clinical advantage over other approaches – as evidenced by DAI, which was similar in other approaches, including a “simple” anti-VCAM coating (for instance the Med Sci Monit 2019 paper). Moreover, other engineering approaches, both genetic and enzymatic, have shown a significantly higher targeting efficiency in other models (the approach by Sackstein et al. Nat Met 2008 is only one example).

Response: We sincerely appreciate the valuable feedback provided by the reviewer. In Supplementary Table 3, we compared the therapeutic effects of various engineered MSCs in colitis models. Although precise comparisons were challenging due to different experimental systems and conditions, our analysis yielded valuable findings. Specifically, the use of polyvalent engineered MSCs offers several advantages in the treatment of colitis, such as promoting the restoration of body weight and colon length.

In response to the reviewer's concerns regarding the similarity in DAI among different approaches, we need to emphasize that the treatment duration was different. In the study involving anti-VCAM1 coated MSCs (Chen, Qianqian, et al. Med Sci Monit 25, 4457-4468 (2019).), colitis mice showed a lower level of DAI after a 13-day treatment. In our study, a 7-day treatment with engineered MSCs led to obvious improvements in all indicators in the PAV-MSCs group. This observation allows us to conclude that PAV-MSCs exhibit enhanced therapeutic effects in colitis treatment. Furthermore, it is important to consider that drawing conclusions solely based on DAI results obtained at varying treatment durations may not be appropriate. The existing researches highlight the importance of considering multiple indicators of recovery in colitis treatment (Zhou, J., et al. Nat Commun 13, 3432 (2022).). Therefore, in addition to DAI, our study systematically assessed additional parameters including changes in body weight, colon length, expression of inflammatory factors and colon tissue damage in colitis mice.

Genetic engineering methods have been employed to enhance the homing efficiency of MSCs, showing promising results in certain animal models. However, direct comparisons of targeting efficiencies between different disease models may not be appropriate due to variations in the models used. In the context of the colitis model, our findings reveal that PAV-MSCs exhibited higher targeting efficiency compared to both MAV-MSCs and the anti-VCAM1 coated MSCs reported in previous studies.

Addition of the Anti-VCAM1 antibody as a control group was important, strengthening previous reports that VCAM-1 antibody itself attenuates colitis (Revised Figure 5). Given the reported importance of PAV vs MAV reported by authors, it would have been appropriate to use such PAV antibodies for this control group, instead of a single monovalent antibody...

Response: We appreciate the great suggestion from the reviewer. As suggested, we have included one more control group using PAV to treat colitis in addition to MAV (Fig. 5 in revised manuscript). The results indicate that while treatment with PAV mildly attenuate colitis, there is no difference in therapeutic effects between PAV and MAV. In comparison, PAV-MSCs can promote tissue repair more effectively. These results suggest that the primary therapeutic impact within the PAV-MSCs construct can be attributed to the MSC component, and the strategic incorporation of PAV modification on the MSC surface enhance the delivery of MSCs to

damaged tissues, consequently yielding a more potent therapeutic effect. The above comprehensive assessment further highlights the pivotal role of surface modification in bolstering the overall therapeutic efficacy of PAV-MSCs.

Fig. 5. d) Changes in the body weight of mice receiving different the treatments within 14 days. e) DAI scores of mice in each group over 14 days. f and g) Quantitative analysis of colon length (f) and the appearance of colons harvested from mice (g) after the different treatments. For d, e and f, Mean \pm SEM, n=5 mice. h) The MPO activity of colons after the different treatments. i-k) The levels of TNF- α , IL-6, and IL-10 in colon tissues after the different treatments. For h-k, Mean \pm SEM, n=5 mice. l) Representative H&E staining images of colon tissue harvested on Day 14 after the different treatments. Representative images out of 7 images obtained are shown. Statistical analysis was performed by one-way ANOVA with Tukey's multiple comparisons tests (*P<0.5; **P<0.01; ***P<0.001; ****P<0.0001; NS, nonsignificant).

Moreover, the biodistribution experiments (important addition!) does not show a significant shift in biodistribution vs other engineering approaches, further highlighting a modest increase in targeting efficiency. The relative instability of this cell modification reported in Sup. Fig.4 and the antibody shedding reported in Sup. Fig.7 also raises concerns regarding this modification approach and its long-term clinical efficacy vs other approaches, such as stable/transient genetic modifications.

Response: We thank the reviewer for the comments. In our experiments using mouse ear inflammation and colitis models, we observed a decrease in the number of PAV-MSCs trapped in the lungs during biodistribution analysis. We interpret this result as a favorable outcome, as it suggests that more PAV-MSCs were able to accumulate in the target organs. In addition, we value the reviewer's feedback and have taken the comments into account, making necessary revisions to the wording of the manuscript. The changes are highlighted in revised manuscript. For instance:

.....The resulting engineered MSCs exhibited **increased** adhesion to vascular endothelial cells in vitro and in vivo and showed **enhanced** therapeutic efficacy in IBD mice.....

We hope that the subsequent explanations will address the concerns raised by the reviewer regarding the stability of this cell modification and its long-term clinical efficacy. Recent studies have demonstrated that the homing process of MSCs to inflamed tissues exhibits similarities with the migration of circulating leukocytes towards inflammatory sites, indicating that it is a transient occurrence (Yuan, M., et al. *Stem Cell Res Ther* 13, 179 (2022)). It is worth noting that, the main goal of our approach is to transiently modulate MSC function by cell surface engineering to enhance its targeted adhesion to vascular endothelial cells, while preserving their own long-term therapeutic functionality. By using this non-genetic engineering approach, we can promote the delivery efficiency of MSCs without compromising their important role of secreting beneficial factors for tissue repair and inflammation regulation. Furthermore, the non-genetic engineering methods offer simpler preparation process and lower cost (Ullah, M., et al. *iScience* 15, 421-438 (2019)). Our findings suggest that the antibodies present on the PAV-MSCs surface exhibit a sufficient retention time, thereby facilitating MSC homing to a significant degree. Once firmly attached to the vascular endothelial cells, MSCs can then migrate across the endothelial layer and reach the damaged tissue to exert their therapeutic effects. Although the antibody shedding occurs during MSC migration, we view it as positive phenomenon because, at this stage, the antibodies have successfully fulfilled their role in promoting MSC adhesion to vascular endothelial cells. In addition, the shedding of these modifiers is unlikely to impact the subsequent therapeutic efficacy of MSCs.

Overall, while the manuscript is now technically improved, one should wonder whether this report of yet another antibody coating approach provides meaningful clinical advantage or makes a substantial contribution to the fields of cell engineering or MSC therapies.

Response: We thank the reviewer for the comments. In this manuscript, we have employed polyvalent engineering approach, which expands beyond the conventional practice of coating a single antibody on the cell surface. Our strategy involves covalently linking antibodies to DNA, enabling the modification of multiple monospecific antibodies on the cell surface through self-assembly. Leveraging the precise programmability of DNA, we have successfully demonstrated the potential of polyvalent anti-VCAM1 to enhance the targeting efficiency of MSCs. Building upon this progress, our research team is currently exploring the assembly of multiple antibodies with different specificities using DNA self-assembly, aiming to further improve MSC targeting efficiency via the synergistic effects of different adhesion molecules. The results obtained thus far strongly support the notion that the DNA self-assembly-based cell engineering method holds significant promise in the field of cell surface engineering. This innovative approach has considerable potential for advancing both MSC therapy and cell engineering. The ability to modify cells at the non-genetic engineering level utilizing genetic materials represents a meaningful advancement in this field. Continued research in this area is expected to yield novel and cutting-edge approaches for cell therapy and engineering.

REVIEWERS' COMMENTS

Reviewer #1 (Remarks to the Author):

The authors have addressed my comments. The manuscript is ready for publication.

Reviewer #1 (Remarks to the Author):

The authors have addressed my comments. The manuscript is ready for publication.

Response: We sincerely appreciate reviewers' recognition of our work and suggestions for improving manuscript quality.